# COMMUNICATION-EFFICIENT AND DRIFT-ROBUST FEDERATED LEARNING VIA ELASTIC NET

## ABSTRACT

Federated learning (FL) is a distributed method to train a global model over a set of local clients while keeping data localized. It reduces the risks of privacy and security but faces important challenges including expensive communication costs and client drift issues. To address these issues, we propose FedElasticNet, a communication-efficient and drift-robust FL framework leveraging the elastic net. It repurposes two types of the elastic net regularizers (i.e., $\ell_1$ and $\ell_2$ penalties on the local model updates): (1) the $\ell_1$-norm regularizer sparsifies the local updates to reduce the communication costs and (2) the $\ell_2$-norm regularizer resolves the client drift problem by limiting the impact of drifting local updates due to data heterogeneity. FedElasticNet is a general framework for FL; hence, without additional costs, it can be integrated into prior FL techniques, e.g., FedAvg, FedProx, SCAFFOLD, and FedDyn. We show that our framework effectively resolves the communication cost and client drift problems simultaneously.

## 1 INTRODUCTION

Federated learning (FL) is a collaborative method that allows many clients to contribute individually to training a global model by sharing local models rather than private data. Each client has a local training dataset, which it does not want to share with the global server. Instead, each client computes an update to the current global model maintained by the server, and only this update is communicated. FL significantly reduces the risks of privacy and security (McMahan et al., 2017; Li et al., 2020a), but it faces crucial challenges that make the federated settings distinct from other classical problems (Li et al., 2020a) such as expensive communication costs and client drift problems due to heterogeneous local training datasets and heterogeneous systems (McMahan et al., 2017; Li et al., 2020a; Konečný et al., 2016a;b).

Communicating models is a critical bottleneck in FL, in particular when the federated network comprises a massive number of devices (Bonawitz et al., 2019; Li et al., 2020a; Konečný et al., 2016b). In such a scenario, communication in the federated network may take a longer time than that of local computation by many orders of magnitude because of limited communication bandwidth and device power (Li et al., 2020a). To reduce such communication cost, several strategies have been proposed (Konečný et al., 2016b; Li et al., 2020a). In particular, Konečný et al. (2016b) proposed several methods to form structured local updates and approximate them, e.g., subsampling and quantization. Reisizadeh et al. (2020); Xu et al. (2020) also proposed an efficient quantization method for FL to reduce the communication cost.

Also, in general, as the datasets that local clients own are heterogeneous, trained models on each local data are *inconsistent* with the global model that minimizes the global empirical loss (Karimireddy et al., 2020; Malinovskiy et al., 2020; Acar et al., 2021). This issue is referred to as the *client drift* problem. In order to resolve the client drift problem, FedProx (Li et al., 2020b) added a proximal term to a local objective function and regulated local model updates. Karimireddy et al. (2020) proposed SCAFFOLD algorithm that transfers both model updates and control variates to resolve the client drift problem. FedDyn (Acar et al., 2021) dynamically regularizes local objective functions to resolve the client drift problem.

Unlike most prior works focusing on either the communication cost problem or the client drift problem, we propose a technique that effectively resolves the communication cost and client drift problems simultaneously.

| | FedAvg | FedProx | SCAFFOLD | FedDyn | **FedElasticNet** |
|---|---|---|---|---|---|
| Communication efficiency | △ | △ | × | △ | ○ |
| Robustness to heterogeneous data | × | △ | ○ | ○ | ○ |

Table 1: Comparison of prior methods and the proposed FedElasticNet.

**Contributions**   In this paper, we propose FedElasticNet, a new framework for communication-efficient and drift-robust FL. It repurposes the $\ell_1$-norm and $\ell_2$-norm regularizers of the elastic net (Zou & Hastie, 2005), by which it successfully improves (i) communication efficiency by adopting the $\ell_1$-norm regularizer and (ii) robustness to heterogeneous local datasets by adopting the $\ell_2$-norm regularizer.

FedElasticNet is a general framework; hence, it can be integrated with prior FL algorithms such as FedAvg (McMahan et al., 2017), FedProx (Li et al., 2020b), SCAFFOLD (Karimireddy et al., 2020), and FedDyn (Acar et al., 2021) so as to resolve the client drift problem as well as the communication cost problem. Further, it incurs no additional costs in training. Empirically, we show that FedElasticNet enhances communication efficiency while maintaining the classification accuracy even for heterogeneous datasets, i.e., the client drift problem is resolved. Theoretically, we characterize the impact of the regularizer terms. Table 1 compares the prior methods and the proposed FedElasticNet if integrated with FedDyn (Algorithm 3).

## 2   RELATED WORK

To address the communication cost and client drift problems, numerous approaches were proposed. Here, we describe closely related works that we consider baseline methods. The comprehensive reviews can be found in Kairouz et al. (2021); Li et al. (2020a).

FedAvg (McMahan et al., 2017) is one of the most commonly used methods. FedAvg tackles the communication bottleneck issue by performing multiple local updates before communicating to the server. It works well for homogeneous datasets across clients (McMahan et al., 2017; Karimireddy et al., 2020), but it is known that FedAvg may diverge when local datasets are heterogeneous (Zhao et al., 2018; Li et al., 2020a).

FedProx (Li et al., 2020b) addressed the data heterogeneity problem. FedProx introduces an $\ell_2$-norm regularizer to the local objective functions to penalize local updates that are far from the server's model and thus to limit the impact of variable local updates (Li et al., 2020b). Although FedProx is more robust to heterogeneous datasets than FedAvg, the regularizer does not result in aligning the global and local stationary points (Acar et al., 2021). Also, we note that FedProx does not improve communication efficiency compared to that of FedAvg.

SCAFFOLD (Karimireddy et al., 2020) defined *client drift* that the model created by aggregating local models and the optimal global model is inconsistent because of heterogeneous local datasets. SCAFFOLD communicates the trained local models and the clients' control variates so as to resolve the client drift problem. Hence, SCAFFOLD requires twice the communication cost compared to other FL algorithms.

FedDyn (Acar et al., 2021) dynamically updates its local regularizers at each round to ensure that the local clients' optima are asymptotically consistent with stationary points of the global empirical loss. Unlike SCAFFOLD, FedDyn resolves the client drift problem without incurring additional communication costs. However, FedDyn's communication cost is not improved compared to FedAvg and FedProx.

Zou & Hastie (2005) proposed the elastic net to encourage the grouping effect, in other words, to encourage strongly correlated covariates to be in or out of the model description together (Hu et al., 2018). Initially, the elastic net was proposed to overcome the limitations of Lasso (Tibshirani, 1996) imposing an $\ell_1$-norm penalty on the model parameters. For instance of a linear least square problem,

the objective of Lasso is to solve

$$\min_{\theta} \|y - \mathbf{X}\theta\|_2^2 + \lambda_1 \|\theta\|_1 , \tag{1}$$

where $y$ is the outcome and $\mathbf{X}$ is the covariate matrix. Lasso performs both variable selection and regularization to enhance the prediction accuracy and interpretability of the resulting model. However, it has some limitations, especially for high-dimensional models. If a group of variables is highly correlated, then Lasso tends to select only one variable from the group and does not care which one is selected (Zou & Hastie, 2005). The elastic net overcomes these limitations by adding an $\ell_2$-norm penalty. The objective of the elastic net is to solve

$$\min_{\theta} \|y - \mathbf{X}\theta\|_2^2 + \frac{\lambda_2}{2} \|\theta\|_2^2 + \lambda_1 \|\theta\|_1 . \tag{2}$$

The elastic net simultaneously enables automatic variable selection and continuous shrinkage by the $\ell_1$-norm regularizer and enables to select groups of correlated variables by its $\ell_2$-norm regularizer (Zou & Hastie, 2005). We will leverage the elastic net approach to resolve the critical problems of FL: expensive communication cost and client drift problems.

## 3 PROPOSED METHOD: FEDELASTICNET

We assume that $m$ local clients communicate with the global server. For the $k$th client (where $k \in [m]$) participating in each training round, we assume that a training data feature $x \in \mathcal{X}$ and its corresponding label $y \in \mathcal{Y}$ are drawn IID from a device-indexed joint distribution, i.e., $(x, y) \sim P_k$ (Acar et al., 2021). The objective is to find

$$\arg\min_{\theta \in \mathbb{R}^d} \left[ \mathcal{R}(\theta) := \frac{1}{m} \sum_{k \in [m]} L_k(\theta) \right] , \tag{3}$$

where $L_k(\theta) = \mathbb{E}_{x \sim P_k} [l_k(\theta; (x, y))]$ is the local risk of the $k$th clients over possibly heterogeneous data distributions $P_k$. Also, $\theta$ represents the model parameters and $l_k(\cdot)$ is a loss function such as cross entropy (Acar et al., 2021).

**FedElasticNet** The proposed method (FedElasticNet) leverages the elastic net approach to resolve the communication cost and client drift problems. We introduce the $\ell_1$-norm and $\ell_2$-norm penalties on the local updates: In each round $t \in [T]$, the $k$th local client attempts to find $\theta_k^t$ by solving the following optimization problem:

$$\theta_k^t = \arg\min_{\theta} L_k(\theta) + \frac{\lambda_2}{2} \|\theta - \theta^{t-1}\|_2^2 + \lambda_1 \|\theta - \theta^{t-1}\|_1 , \tag{4}$$

where $\theta^{t-1}$ denotes the global model received from the server. Then, it transmits the difference $\Delta_k^t = \theta_k^t - \theta^{t-1}$ to the server.

Inspired by the elastic net, we introduce two types of regularizers for local objective functions; however, each of them works in a different way so as to resolve each of the two FL problems: the communication cost and client drift problems. First, the $\ell_2$-norm regularizer resolves the client drift problem by limiting the impact of variable local updates as in FedProx (Li et al., 2020b). FedDyn (Acar et al., 2021) also adopts the $\ell_2$-norm regularizer to control the client drift.

Second, the $\ell_1$-norm regularizer attempts to sparsify the local updates $\Delta_k^t = \theta_k^t - \theta^{t-1}$. We consider two ways of measuring communication cost: One is the number of nonzero elements in $\Delta_k^t$ (Yoon et al., 2021; Jeong et al., 2021), which the $\ell_1$-norm sparsifies. The other is the (Shannon) entropy since it is the theoretical lower bound on the data compression (Cover & Thomas, 2006). We demonstrate that the $\ell_1$-norm penalty on the local updates can effectively reduce the number of nonzero elements as well as the entropy in Section 4. To boost sparseness of $\Delta_k^t = \theta_k^t - \theta^{t-1}$, we sent $\Delta_k^t(i) = 0$ if $|\Delta_k^t(i)| \leq \epsilon$ where $\Delta_k^t(i)$ denotes the $i$th element of $\Delta_k^t$. The parameter $\epsilon$ is chosen in a range that does not affect classification accuracy.

Our FedElasticNet approach can be integrated into existing FL algorithms such as FedAvg (McMahan et al., 2017), SCAFFOLD (Karimireddy et al., 2020), and FedDyn (Acar et al., 2021) without additional costs, which will be described in the following subsections.

---

**Algorithm 1** FedElasticNet for FedAvg & FedProx

---

**Input:** $T, \theta^0, \lambda_1 > 0, \lambda_2 > 0$
1: **for** each round $t = 1, 2, ..., T$ **do**
2:      Sample devices $\mathcal{P}_t \subseteq [m]$ and transmit $\theta^{t-1}$ to each selected local client
3:      **for** each local client $k \in \mathcal{P}_t$ **do in parallel**
4:          Set $\theta_k^t = \arg\min_\theta L_k(\theta) + \frac{\lambda_2}{2}\left\|\theta - \theta^{t-1}\right\|_2^2 + \lambda_1 \left\|\theta - \theta^{t-1}\right\|_1$
5:          Transmit $\Delta_k^t = \theta_k^t - \theta^{t-1}$ to the global server
6:      **end for**
7:      Set $\theta^t = \theta^{t-1} + \sum_{k \in \mathcal{P}_t} \frac{n_k}{n} \Delta_k$
8: **end for**

---

**Algorithm 2** FedElasticNet for SCAFFOLD

---

**Input:** $T, \theta^0, \lambda_1 > 0, \lambda_2 > 0$, global step size $\eta_g$, and local step size $\eta_l$.
1: **for** each round $t = 1, 2, ..., T$ **do**
2:      Sample devices $\mathcal{P}_t \subseteq [m]$ and transmit $\theta^{t-1}$ and $c^{t-1}$ to each selected device
3:      **for** each device $k \in \mathcal{P}_t$ **do in parallel**
4:          Initialize local model $\theta_k^t = \theta^{t-1}$
5:          **for** $b = 1, \ldots, B$ **do**
6:              Compute mini-batch gradient $\nabla L_k(\theta_k^t)$
7:              $\theta_k^t \leftarrow \theta_k^t - \eta_l\left(\nabla L_k(\theta_k^t) - c_k^{t-1} + c^{t-1} + \lambda_2(\theta_k^t - \theta^{t-1}) + \lambda_1 \mathrm{sign}(\theta_k^t - \theta^{t-1})\right)$
8:          **end for**
9:          Set $c_k^t = c_k^{t-1} - c^{t-1} + \frac{1}{B\eta_l}(\theta^{t-1} - \theta_k^t)$
10:        Transmit $\Delta_k^t = \theta_k^t - \theta^{t-1}$ and $\Delta c_k = c_k^t - c_k^{t-1}$ to the global server
11:      **end for**
12:      Set $\theta^t = \theta^{t-1} + \frac{\eta_g}{|\mathcal{P}_t|}\sum_{k \in \mathcal{P}_t}\Delta_k$
13:      Set $c^t = c^{t-1} + \frac{1}{m}\sum_{k \in \mathcal{P}_t}\Delta c_k$
14: **end for**

---

## 3.1 FedElasticNet for FedAvg & FedProx (FedAvg & FedProx + Elastic Net)

Our FedElasticNet can be applied to FedAvg (McMahan et al., 2017) by adding two regularizers on the local updates, which resolves the client drift problem and the communication cost problem. As shown in Algorithm 1, the local client minimizes the local objective function (4). In Step 7, $n$ and $n_k$ denote the total numbers of data points of all clients and the data points of the $k$th client, respectively.

It is worth mentioning that FedProx uses the $\ell_2$-norm regularizer to address the data and system heterogeneities (Li et al., 2020b). By adding the $\ell_1$-norm regularizer, we can sparsify the local updates of FedProx and thus effectively reduce the communication cost. Notice that Algorithm 1 can be viewed as the integration of FedProx and FedElasticNet.

## 3.2 FedElasticNet for SCAFFOLD (SCAFFOLD + Elastic Net)

In SCAFFOLD, each client computes the following mini-batch gradient $\nabla L_k(\theta_k^t)$ and control variate $c_k^t$ (Karimireddy et al., 2020):

$$\theta_k^t \leftarrow \theta_k^t - \eta_l\left(\nabla L_k(\theta_k^t) - c_k^{t-1} + c^{t-1}\right), \tag{5}$$

$$c_k^t \leftarrow c_k^{t-1} - c^{t-1} + \frac{1}{B\eta_l}(\theta^{t-1} - \theta_k^t), \tag{6}$$

where $\eta_l$ is the local step size and $B$ is the number of mini-batches at each round. This control variate makes the local parameters $\theta_k^t$ updated in the direction of the global optimum rather than each local optimum, which effectively resolves the client drift problem. However, SCAFFOLD incurs twice much communication cost since it should communicate the local update $\Delta_k^t = \theta_k^t - \theta^{t-1}$ and the control variate $\Delta c_k = c_k^t - c_k^{t-1}$, which are of the same dimension.

In order to reduce the communication cost of SCAFFOLD, we apply our FedElasticNet framework. In the proposed algorithm (see Algorithm 2), each local client computes the following mini-batch

gradient instead of (5):

$$\theta_k^t \leftarrow \theta_k^t - \eta_l \left( \nabla L_k \left( \theta_k^t \right) - c_k^{t-1} + c^{t-1} + \lambda_2(\theta_k^t - \theta^{t-1}) + \lambda_1 \text{sign}(\theta_k^t - \theta^{t-1}) \right), \quad (7)$$

where $\lambda_1 \text{sign}(\theta_k^t - \theta^{t-1})$ corresponds to the gradient of $\ell_1$-norm regularizer $\lambda_1 \|\theta_k^t - \theta^{t-1}\|_1$. This $\ell_1$-norm regularizer sparsifies the local update $\Delta_k^t = \theta_k^t - \theta^{t-1}$; hence, reduces the communication cost. Since the control variate already addresses the client drift problem, we can remove the $\ell_2$-norm regularizer or set $\lambda_2$ as a small value.

### 3.3 FedElasticNet for FedDyn (FedDyn + Elastic Net)

In FedDyn, each local client optimizes the following local objective, which is the sum of its empirical loss and a penalized risk function:

$$\theta_k^t = \arg \min_\theta L_k(\theta) - \langle \nabla L_k(\theta_k^{t-1}), \theta \rangle + \frac{\lambda_2}{2} \left\| \theta - \theta^{t-1} \right\|_2^2, \quad (8)$$

where the penalized risk is dynamically updated so as to satisfy the following first-order condition for local optima:

$$\nabla L_k(\theta_k^t) - \nabla L_k(\theta_k^{t-1}) + \lambda_2(\theta_k^t - \theta^{t-1}) = 0. \quad (9)$$

This first-order condition shows that the stationary points of the local objective function are consistent with the server model (Acar et al., 2021). That is, the client drift is resolved. However, FedDyn makes no difference from FedAvg and FedProx in communication costs.

By integrating FedElasticNet and FedDyn, we can effectively reduce the communication cost of FedDyn as well. In the proposed method (i.e., FedElasticNet for FedDyn), each local client optimizes the following local empirical objective:

$$\theta_k^t = \arg \min_\theta L_k(\theta) - \langle \nabla L_k(\theta_k^{t-1}), \theta \rangle + \frac{\lambda_2}{2} \left\| \theta - \theta^{t-1} \right\|_2^2 + \lambda_1 \left\| \theta - \theta^{t-1} \right\|_1, \quad (10)$$

which is the sum of (8) and the additional $\ell_1$-norm penalty on the local updates. The corresponding first-order condition is given by

$$\nabla L_k(\theta_k^t) - \nabla L_k(\theta_k^{t-1}) + \lambda_2(\theta_k^t - \theta^{t-1}) + \lambda_1 \text{sign}(\theta_k^t - \theta^{t-1}) = 0. \quad (11)$$

Notice that the stationary points of the local objective function are consistent with the server model as in (9). If $\theta_k^t \neq \theta^{t-1}$ (i.e., $\text{sign}(\theta_k^t - \theta^{t-1}) = \pm 1$), then the first-order condition is

$$\nabla L_k(\theta_k^t) - \nabla L_k(\theta_k^{t-1}) + \lambda_2(\theta_k^t - \theta^{t-1}) = \pm \lambda_1, \quad (12)$$

where $\lambda_1$ is a vectorized one. Our empirical results show that the optimized hyperparameter is $\lambda_1 = 10^{-4}$ or $10^{-6}$ and the impact of $\pm \lambda_1$ in (12) would be negligible. Hence, the proposed FedElasticNet for FedDyn resolves the client drift problem. Further, the local update $\Delta_k^t = \theta_k^t - \theta^{t-1}$ is sparse due to the $\ell_1$-norm regularizer, which effectively reduces the communication cost at the same time. The detailed algorithm is described in Algorithm 3.

---

**Algorithm 3** FedElasticNet for FedDyn

---

**Input:** $T, \theta^0, \lambda_1 > 0, \lambda_2 > 0, h^0 = 0, \nabla L_k(\theta_k^0) = 0$.
1: **for** each round $t = 1, 2, ..., T$ **do**
2:     Sample devices $\mathcal{P}_t \subseteq [m]$ and transmit $\theta^{t-1}$ to each selected device
3:     **for** each device $k \in \mathcal{P}_t$ **do in parallel**
4:         Set $\theta_k^t = \arg \min_\theta L_k(\theta) - \langle \nabla L_k(\theta_k^{t-1}), \theta \rangle + \frac{\lambda_2}{2} \left\| \theta - \theta^{t-1} \right\|_2^2 + \lambda_1 \left\| \theta - \theta^{t-1} \right\|_1$
5:         Set $\nabla L_k(\theta_k^t) = \nabla L_k(\theta_k^{t-1}) - \lambda_2(\theta_k^t - \theta^{t-1}) - \lambda_1 \text{sign}(\theta_k^t - \theta^{t-1})$
6:         Transmit $\Delta_k^t = \theta_k^t - \theta^{t-1}$ to the global server
7:     **end for**
8:     **for** each device $k \notin \mathcal{P}_t$ **do in parallel**
9:         Set $\theta_k^t = \theta_k^{t-1}$ and $\nabla L_k(\theta_k^t) = \nabla L_k(\theta_k^{t-1})$
10:     **end for**
11:     Set $h^t = h^{t-1} - \frac{\lambda_2}{m} \sum_{k \in \mathcal{P}_t} (\theta_k^t - \theta^{t-1}) - \frac{\lambda_1}{m} \sum_{k \in \mathcal{P}_t} \text{sign}(\theta_k^t - \theta^{t-1})$
12:     Set $\theta^t = \frac{1}{|\mathcal{P}_t|} \sum_{k \in \mathcal{P}_t} \theta_k^t - \frac{1}{\lambda_2} h^t$
13: **end for**

---

**Convergence Analysis** We provide a convergence analysis on FedElasticNet for FedDyn (Algorithm 3).

**Theorem 3.1.** *Assume that the clients are uniformly randomly selected at each round and the local loss functions are convex and $\beta$-smooth. Then Algorithm 3 satisfies the following inequality:*

$$\mathbb{E}\left[\mathcal{R}\left(\frac{1}{T}\sum_{t=0}^{T-1}\gamma^t\right) - \mathcal{R}(\theta_*)\right] \leq \frac{1}{T}\frac{1}{\kappa_0}(\mathbb{E}\|\gamma^0 - \theta_*\|_2^2 + \kappa C_0) + \frac{\kappa'}{\kappa_0}\cdot\lambda_1^2 d$$

$$- \frac{1}{T}\frac{2\lambda_1}{\lambda_2}\sum_{t=1}^{T}\left\langle\gamma^{t-1} - \theta_*, \frac{1}{m}\sum_{k\in[m]}\mathbb{E}[\text{sign}(\tilde{\theta}_k^t - \theta^{t-1})]\right\rangle, \quad (13)$$

*where $\theta_* = \arg\min_\theta \mathcal{R}(\theta)$, $P = |\mathcal{P}_t|$, $\gamma^t = \frac{1}{P}\sum_{\mathcal{P}_t}\theta_k^t$, $d = \dim(\theta)$, $\kappa = \frac{10m}{P}\frac{1}{\lambda_2}\frac{\lambda_2+\beta}{\lambda_2^2-25\beta^2}, \kappa_0 = \frac{2}{\lambda_2}\frac{\lambda_2^2-25\lambda_2\beta-50\beta^2}{\lambda_2^2-25\beta^2}, \kappa' = \frac{5}{\lambda_2}\frac{\lambda_2+\beta}{\lambda_2^2-25\beta^2} = \kappa\cdot\frac{P}{2m}, C_0 = \frac{1}{m}\sum_{k\in[m]}\mathbb{E}\|\nabla L_k(\theta_k^0) - \nabla L_k(\theta_*)\|$ and*

$$\tilde{\theta}_k^t = \arg\min_\theta L_k(\theta) - \langle\nabla L_k(\theta_k^{t-1}), \theta\rangle + \frac{\lambda_2}{2}\|\theta - \theta^{t-1}\|_2^2 + \lambda_1\|\theta - \theta^{t-1}\|_1 \quad \forall k \in [m].$$

Theorem 3.1 provides a convergence rate of FedElasticNet for FedDyn. If $T \to \infty$, the first term of (13) converges to 0 at the speed of $\mathcal{O}(1/T)$. The second and the third terms of (13) are additional penalty terms caused by the $\ell_1$-norm regularizer. The second term is a negligible constant in the range of hyperparameters of our interest. Considering the last term, notice that the summand at each $t$ includes the expected average of sign vectors where each element is $\pm 1$. If a coordinate of the sign vectors across clients is viewed as an IID realization of $\text{Bern}(\frac{1}{2})$, it can be thought of as a small value with high probability by the concentration property (see Appendix B.3). In addition, $\gamma^{t-1} - \theta_*$ characterizes how much the average of local models deviates from the globally optimal model, which tends to be small as training proceeds. Therefore, the effect of both additional terms is negligible.

# 4 EXPERIMENTS

In this section, we evaluate the proposed FedElasticNet on benchmark datasets for various FL scenarios. In particular, FedElasticNet is integrated with prior methods including FedProx (Li et al., 2020b), SCAFFOLD (Karimireddy et al., 2020), and FedDyn (Acar et al., 2021). The experimental results show that FedElasticNet effectively enhances communication efficiency while maintaining classification accuracy and resolving the client drift problem. We observe that the integration of FedElasticNet and FedDyn (Algorithm 3) achieves the best performance.

**Experimental Setup** We use the same benchmark datasets as prior works. The evaluated datasets include MNIST (LeCun et al., 1998), a subset of EMNIST (Cohen et al., 2017, EMNIST-L), CIFAR-10, CIFAR-100 (Krizhevsky & Hinton, 2009), and Shakespeare (Shakespeare, 1914). The IID split is generated by randomly assigning datapoint to the local clients. The Dirichlet distribution is used on the label ratios to ensure uneven label distributions among local clients for non-IID splits as in Zhao et al. (2018); Acar et al. (2021). For the uneven label distributions among 100 experimental devices, the experiments are performed by using the Dirichlet parameters of 0.3 and 0.6, and the number of data points is obtained by the lognormal distribution as in Acar et al. (2021). The data imbalance is controlled by varying the variance of the lognormal distribution (Acar et al., 2021).

We use the same neural network models of FedDyn experiments (Acar et al., 2021). For MNIST and EMNIST-L, fully connected neural network architectures with 2 hidden layers are used. The numbers of neurons in the layers are 200 and 100, respectively (Acar et al., 2021). Remark that the model used for MNIST dataset is the same as in Acar et al. (2021); McMahan et al. (2017). For CIFAR-10 and CIFAR-100 datasets, we use a CNN model consisting of 2 convolutional layers with 64 $5 \times 5$ filters followed by 2 fully connected layers with 394 and 192 neurons and a softmax layer. For the next character prediction task for Shakespeare, we use a stacked LSTM as in Acar et al. (2021).

For MNIST, EMNIST-L, CIFAR10, and CIFAR100 datasets, we evaluate three cases: IID, non-IID with Dirichlet (.6), and non-IID with Dirichlet (.3). Shakespeare datasets are evaluated for IID and non-IID cases as in Acar et al. (2021). We use the batch size of 10 for the MNIST dataset, 50 for CIFAR-10, CIFAR-100, and EMNIST-L datasets, and 20 for the Shakespeare dataset. We optimize the hyperparameters depending on the evaluated datasets: learning rates, $\lambda_2$, and $\lambda_1$.

| | Dataset | Rounds | FedProx | Algorithm 1 | SCAFFOLD | Algorithm 2 | FedDyn | Algorithm 3 |
|---|---|---|---|---|---|---|---|---|
| **IID** | CIFAR-10 | 200 | 163.82 | **124.56** | 327.64 | **313.04** | 163.82 | **34.23** |
| | CIFAR-100 | 500 | 413.37 | **249.26** | 826.74 | **803.33** | 413.37 | **132.95** |
| | MNIST | 100 | 20.3 | **7.51** | 40.58 | **36.98** | 20.3 | **2.55** |
| | EMNIST-L | 200 | 18.4 | **11.23** | 36.78 | **34.90** | 18.4 | **2.12** |
| | Shakespeare | 100 | 1.99 | **1.93** | 3.32 | **3.31** | 1.99 | **1.28** |
| **Dirichlet (.6)** | CIFAR-10 | 200 | 163.82 | **116.96** | 327.64 | **310.51** | 163.82 | **29.96** |
| | CIFAR-100 | 500 | 413.37 | **247.87** | 826.74 | **798.21** | 413.37 | **127.23** |
| | MNIST | 100 | 20.3 | **6.66** | 40.58 | **35.20** | 20.3 | **2.55** |
| | EMNIST-L | 200 | 18.4 | **11.18** | 36.78 | **34.45** | 18.4 | **2.05** |
| **Dirichlet (.3)** | CIFAR-10 | 200 | 163.82 | **112.1** | 327.64 | **308.15** | 163.82 | **26.99** |
| | CIFAR-100 | 500 | 413.37 | **255.97** | 826.74 | **804.47** | 413.37 | **124.02** |
| | MNIST | 100 | 20.3 | **6.49** | 40.58 | **34.95** | 20.3 | **2.53** |
| | EMNIST-L | 200 | 18.4 | **11.29** | 36.78 | **34.22** | 18.4 | **2.07** |
| **Non-IID** | Shakespeare | 100 | 13.61 | **11.28** | 27.21 | **27.21** | 13.61 | **9.41** |

Table 2: Number of non-zero elements cumulated over the all round simulated with 10% client participation for IID and non-IID settings in FL scenarios. The non-IID settings of MNIST, EMNIST-L, CIFAR-10, and CIFAR-100 datasets are created with the Dirichlet distribution of labels owned by the client. Algorithm 1 is FedElasticNet for FedProx, Algorithm 2 is FedElasticNet for SCAFFOLD, and Algorithm 3 is FedElasticNet for FedDyn. The unit of the cumulative number of elements is $10^7$.

| | Dataset | Rounds | FedProx | Algorithm 1 | SCAFFOLD | Algorithm 2 | FedDyn | Algorithm 3 |
|---|---|---|---|---|---|---|---|---|
| **IID** | CIFAR-10 | 200 | 586.42 | **232.77** | 685.17 | **236.76** | 639.59 (221.64) | **140.71** |
| | CIFAR-100 | 500 | 1712.84 | **470.78** | 2225.98 | **1173.01** | 1964.63 (511.14) | **423.53** |
| | MNIST | 100 | 266.26 | **47.29** | 286.88 | **83.51** | 308.27 (27.76) | **24.76** |
| | EMNIST-L | 200 | 657.64 | **166.07** | 764.39 | **344.94** | 704.57 (132.50) | **96.37** |
| | Shakespeare | 100 | 646.33 | **403.63** | 520.60 | **226.68** | 576.17 (348.11) | **225.44** |
| **Dirichlet (.6)** | CIFAR-10 | 200 | 564.57 | **203.61** | 663.23 | **198.53** | 616.69 (197.62) | **121.97** |
| | CIFAR-100 | 500 | 1709.33 | **449.59** | 2202.61 | **1119.91** | 1951.06 (478.60) | **398.87** |
| | MNIST | 100 | 249.63 | **45.51** | 293.22 | **75.57** | 304.00 (26.75) | **21.24** |
| | EMNIST-L | 200 | 646.14 | **163.24** | 755.75 | **347.31** | 704.63 (134.31) | **89.92** |
| **Dirichlet (.3)** | CIFAR-10 | 200 | 550.15 | **187.01** | 636.90 | **115.26** | 602.80 (180.29) | **108.69** |
| | CIFAR-100 | 500 | 1696.47 | **428.77** | 2170.14 | **1078.97** | 1937.09 (463.67) | **382.44** |
| | MNIST | 100 | 244.49 | **45.24** | 291.88 | **73.12** | 300.76 (26.71) | **19.03** |
| | EMNIST-L | 200 | 636.21 | **162.57** | 747.72 | **328.21** | 700.38 (128.34) | **91.55** |
| **Non-IID** | Shakespeare | 100 | 593.21 | **440.97** | 628.32 | **470.22** | 609.11 (348.11) | **419.99** |

Table 3: Cumulative entropy values of transmitted bits with 10% client participation for IID and non-IID settings in FL scenarios. The non-IID settings of MNIST, EMNIST-L, CIFAR-10, and CIFAR-100 datasets are created with the Dirichlet distribution of labels owned by the client. Algorithm 1 is FedElasticNet for FedProx, Algorithm 2 is FedElasticNet for SCAFFOLD, and Algorithm 3 is FedElasticNet for FedDyn. The left-side numbers of FedDyn are the entropy values when the local models $\theta_k^t$ are transmitted and the right-side numbers in parentheses are the entropy values when the local updates $\Delta_k^t = \theta_k^t - \theta^{t-1}$ are transmitted.

**Evaluation of Methods** We compare the baseline methods (FedProx, SCAFFOLD, and FedDyn) and the proposed FedElasticNet integrations (Algorithms 1, 2, and 3), respectively. We evaluate the communication cost and classification accuracy for non-IID settings of the prior methods and the proposed methods. The robustness of the client drift problem is measured by the classification accuracy of non-IID settings.

We report the communication costs in two ways: (i) the number of nonzero elements in transmitted values as in (Yoon et al., 2021; Jeong et al., 2021) and (ii) the Shannon entropy of transmitted bits. Note that the Shannon entropy is the theoretical limit of data compression (Cover & Thomas, 2006), which can be achieved by practical algorithms; for instance, Han et al. (2016) used Huffman coding for model compression. We calculate the entropy of discretized values with the bin size of 0.01. Note that the transmitted values are not discretized in FL, and only the discretization is considered to calculate the entropy. The lossy compression schemes (e.g., scalar quantization, vector quantization, etc.) have not been considered since they include several implementational issues which are beyond our research scope.

Table 2 reports the number of non-zero elements of the baseline methods with/without FedElasticNet. Basically, the communication costs per round of FedProx and FedDyn are the same; SCAFFOLD suffers from the doubled communication cost because of the control variates. The proposed FedElasticNet integrations (Algorithms 1, 2, and 3) can effectively sparsify the transmitted local updates, which enhances communication efficiency.

| | Dataset | Rounds | FedProx | Algorithm 1 | SCAFFOLD | Algorithm 2 | FedDyn | Algorithm 3 |
|---|---|---|---|---|---|---|---|---|
| **IID** | CIFAR-10 | 200 | 595.16 | **151.28** | 873.24 | **252.33** | 680.70 (259.31) | **57.70** |
| | CIFAR-100 | 500 | 1721.53 | **466.21** | 2774.60 | **1068.47** | 2038.07 (689.24) | **447.18** |
| | MNIST | 100 | 325.86 | **39.55** | 389.52 | **71.57** | 324.88 (20.47) | **12.33** |
| | EMNIST-L | 200 | 728.34 | **123.74** | 1018.11 | **263.95** | 797.39 (55.46) | **40.71** |
| | Shakespeare | 100 | 640.69 | **279.51** | 529.05 | **476.81** | 343.32 (298.58) | **277.21** |
| **Dirichlet (.6)** | CIFAR-10 | 200 | 577.74 | **127.14** | 839.41 | **236.48** | 656.48 (223.46) | **47.52** |
| | CIFAR-100 | 500 | 1697.18 | **453.87** | 2682.15 | **1038.21** | 1974.65 (651.78) | **431.54** |
| | MNIST | 100 | 298.99 | **30.94** | 530.72 | **106.08** | 314.64 (20.29) | **11.67** |
| | EMNIST-L | 200 | 721.49 | **121.76** | 1020.32 | **251.65** | 779.44 (286.11) | **40.71** |
| **Dirichlet (.3)** | CIFAR-10 | 200 | 563.15 | **105.89** | 806.73 | **214.31** | 635.78 (215.83) | **39.58** |
| | CIFAR-100 | 500 | 1685.30 | **444.32** | 2743.43 | **1060.83** | 1934.49 (627.56) | **422.46** |
| | MNIST | 100 | 295.80 | **42.94** | 466.55 | **85.06** | 314.96 (19.98) | **12.35** |
| | EMNIST-L | 200 | 716.12 | **116.88** | 1014.08 | **249.68** | 771.90 (283.40) | **40.70** |
| **Non-IID** | Shakespeare | 100 | 595.69 | **409.60** | 684.79 | 897.67 | 560.87 (316.64) | **318.92** |

Table 4: Cumulative entropy values of transmitted bits with 100% client participation for IID and non-IID settings in FL scenarios. The non-IID settings of MNIST, EMNIST-L, CIFAR-10, and CIFAR-100 datasets are created with the Dirichlet distribution of labels owned by the client. Algorithm 1 is FedElasticNet for FedProx, Algorithm 2 is FedElasticNet for SCAFFOLD, and Algorithm 3 is FedElasticNet for FedDyn. The left-side numbers of FedDyn are the entropy values when the local models $\theta_k^t$ are transmitted and the right-side numbers in parentheses are the entropy values when the local updates $\Delta_k^t = \theta_k^t - \theta^{t-1}$ are transmitted.

In particular, the minimal communication cost is achieved when FedElasticNet is integrated with FedDyn (Algorithm 3). It is because the classification accuracy is not degraded even if the transmitted values are more aggressively sparsified in Algorithm 3. Fig. 2 shows the transmitted local updates $\Delta_k^t$ of Algorithm 3 are sparser than FedDyn and Algorithm 2. Hence, Algorithm 3 (FedElasticNet for FedDyn) achieves the best communication efficiency.

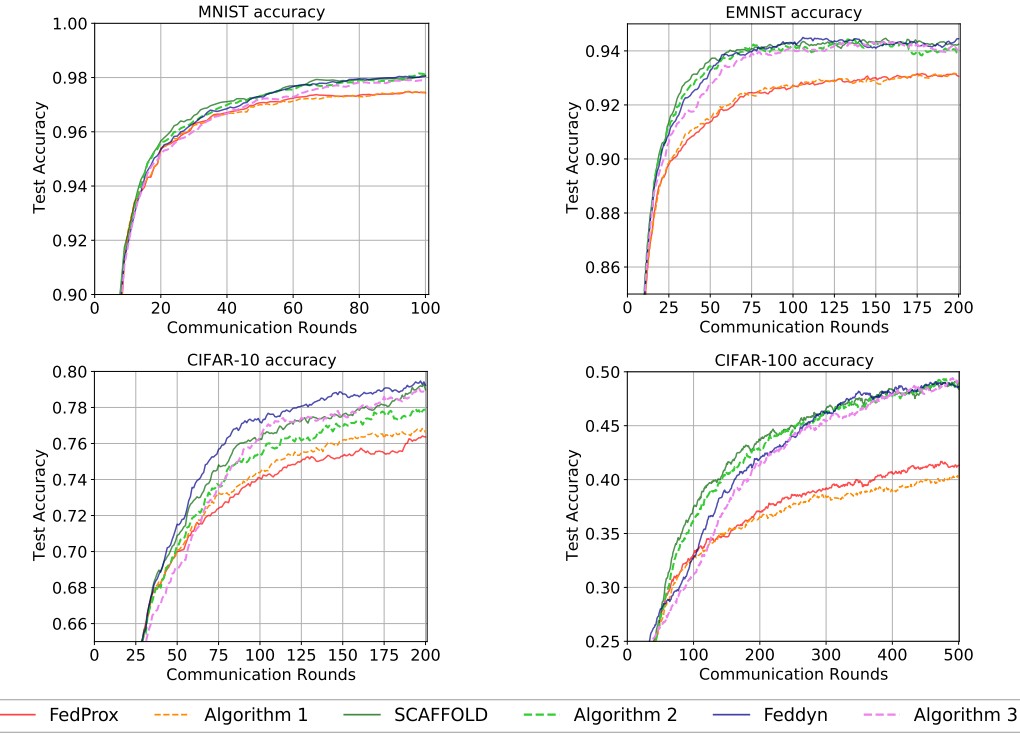

Figure 1: Classification accuracy performance evaluated in MNIST, EMNIST-L, CIFAR-10, CIFAR-100 dataset settings (10% participation rate and Dirichlet (.3)).

Tables 3 and 4 report the Shannon entropy of transmitted bits for the baseline methods with/without FedElasticNet. The communication costs of baseline methods are effectively improved by the

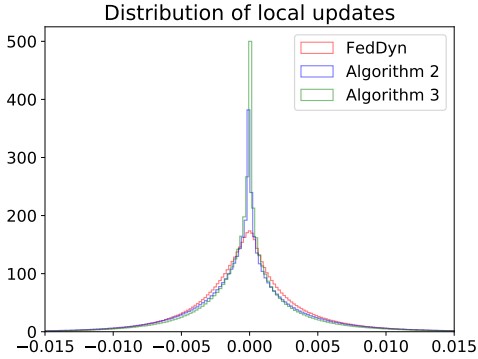

Figure 2: Comparison of distributions of transmitted local updates $\Delta_k^t = \theta_k^t - \theta^{t-1}$ (10% participation rate and Dirichlet (.3)) for CIFAR-10.

FedElasticNet approach. Algorithms 1, 2, and 3 reduce the entropy compared to the their baseline methods. We note that FedElasticNet integrated with FedDyn (Algorithm 3) achieves the minimum entropy, i.e., the minimum communication cost.

For FedDyn, we evaluate the Shannon entropy values for two cases: (i) transmit the updated local models $\theta_k^t$ as in Acar et al. (2021) and (ii) transmit the local updates $\Delta_k^t = \theta_k^t - \theta^{t-1}$ as in Algorithm 3. We observe that transmitting the local updates $\Delta_k^t$ instead of the local models $\theta_k^t$ can reduce the Shannon entropy significantly. Hence, it is beneficial to transmit the local updates $\Delta_k^t$ even for FedDyn if it adopts an additional compression scheme. The numbers of nonzero elements for two cases (i.e., $\theta_k^t$ and $\Delta_k^t$) are the same for FedDyn.

Fig. 1 shows that the FedElasticNet maintains the classification accuracy or incurs marginal degradation. We observe a classification gap between FedProx and Algorithm 1 for CIFAR-10 and CIFAR-100. However, the classification accuracies of FedDyn and Algorithm 3 are almost identical in the converged regime.

In particular, Algorithm 3 significantly reduces the Shannon entropy, which can be explained by Fig 2. Fig 2 compares the distributions of the transmitted local updates $\Delta_k^t$ for FedDyn, Algorithm 2, and Algorithm 3. Because of the $\ell_1$-norm penalty on the local updates, Algorithm 3 makes sparser local updates than FedDyn. The local updates of FedDyn can be modeled by the Gaussian distribution, and the local updates of FedElasticNet can be modeled by the non-Gaussian distribution (similar to the Laplacian distribution). It is well-known that the Gaussian distribution maximizes the entropy for a given variance in information theory Cover & Thomas (2006). Hence, FedElasticNet can reduce the entropy by transforming the Gaussian distribution into the non-Gaussian one.

## 5 CONCLUSION

We proposed FedElasticNet, a general framework to improve communication efficiency and resolve the client drift problem simultaneously. We introduce two types of penalty terms on the local model updates by repurposing the classical elastic net. The $\ell_1$-norm regularizer sparsifies the local model updates, which reduces the communication cost. The $\ell_2$-norm regularizer limits the impact of variable local updates to resolve the client drift problem. Importantly, our framework can be integrated with prior FL techniques so as to simultaneously resolve the communication cost problem and the client drift problem. By integrating FedElasticNet with FedDyn, we can achieve the best communication efficiency while maintaining classification accuracy for heterogeneous datasets.

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

# A APPENDIX

## A.1 EXPERIMENT DETAILS

We provide the details of our experiments. We select the datasets for our experiments, including those used in prior work on federated learning (McMahan et al., 2017; Li et al., 2020b; Acar et al., 2021). To fairly compare the non-IID environments, the datasets and the experimental environments are the same as those of Acar et al. (2021).

**Hyperparameters.** We describe the hyperparameters used in our experiments in Section 4. We perform a grid search to find the best $\lambda_1$ and $\epsilon$ used in the proposed algorithms. Each hyperparameter was selected to double the value as the performance improved. We use the same $\lambda_2$ as in Acar et al. (2021). SCAFFOLD has the same local epoch and batch size as other algorithms, and SCAFFOLD is not included in Table 4 because other hyperparameters are not required. Table 5 shows the hyperparameters used in our experiments.

| Dataset | Algorithm | $\lambda_1$ | $\lambda_2$ | $\epsilon$ |
|---|---|---|---|---|
| **CIFAR-10** | FedProx | - | $10^{-4}$ | - |
| | Algorithm 1 | $10^{-6}$ | $10^{-4}$ | $10^{-3}$ |
| | Algorithm 2 | $10^{-4}$ | 0 | $10^{-4}$ |
| | FedDyn | - | $10^{-2}$ | - |
| | Algorithm 3 | $10^{-4}$ | $10^{-2}$ | $5 \times 10^{-3}$ |
| **CIFAR-100** | FedProx | - | $10^{-4}$ | - |
| | Algorithm 1 | $10^{-6}$ | $10^{-4}$ | $10^{-3}$ |
| | Algorithm 2 | $10^{-4}$ | 0 | $10^{-4}$ |
| | FedDyn | - | $10^{-2}$ | - |
| | Algorithm 3 | $10^{-4}$ | $10^{-2}$ | $10^{-3}$ |
| **MNIST** | FedProx | - | $10^{-4}$ | - |
| | Algorithm 1 | $10^{-6}$ | $10^{-6}$ | $10^{-3}$ |
| | Algorithm 2 | $10^{-4}$ | 0 | $10^{-4}$ |
| | FedDyn | - | $5 \times 10^{-2}$ | - |
| | Algorithm 3 | $10^{-4}$ | $5 \times 10^{-2}$ | $5 \times 10^{-3}$ |
| **EMNIST-L** | FedProx | - | $10^{-4}$ | - |
| | Algorithm 1 | $10^{-6}$ | $10^{-6}$ | $10^{-3}$ |
| | Algorithm 2 | $10^{-4}$ | 0 | $10^{-4}$ |
| | FedDyn | - | $4 \times 10^{-2}$ | - |
| | Algorithm 3 | $10^{-4}$ | $4 \times 10^{-2}$ | $2 \times 10^{-3}$ |
| **Shakespeare** | FedProx | - | $10^{-4}$ | - |
| | Algorithm 1 | $10^{-6}$ | $10^{-6}$ | $9 \times 10^{-3}$ |
| | Algorithm 2 | $10^{-6}$ | 0 | $9 \times 10^{-4}$ |
| | FedDyn | - | $10^{-2}$ | - |
| | Algorithm 3 | $10^{-6}$ | $10^{-2}$ | $10^{-2}$ |

Table 5: Hyperparameters.

## A.2 REGULARIZER COEFFICIENTS

We selected $\lambda_1$ over $\{10^{-2}, 10^{-4}, 10^{-6}, 10^{-8}\}$ to observe the impact of $\lambda_1$ on the classification accuracy. We prefer a larger $\lambda_1$ to enhance communication efficiency unless the $\ell_1$-norm regularizer does not degrade the classification accuracy. Figures 3, 4, and 5 show the classification accuracy depending on $\lambda_1$ in the CIFAR-10 dataset with 10% participation rate and Dirichlet (.3). The unit of the cumulative number of elements is $10^7$.

In Algorithm 1, we selected $\lambda_1 = 10^{-6}$ to avoid a degradation of classification accuracy (see Fig. 3) and maximize the sparsity of local updates. In this way, we selected the coefficient values $\lambda_1$ (See Fig.4 for Algorithm 2 and 5 and Algorithm 3).

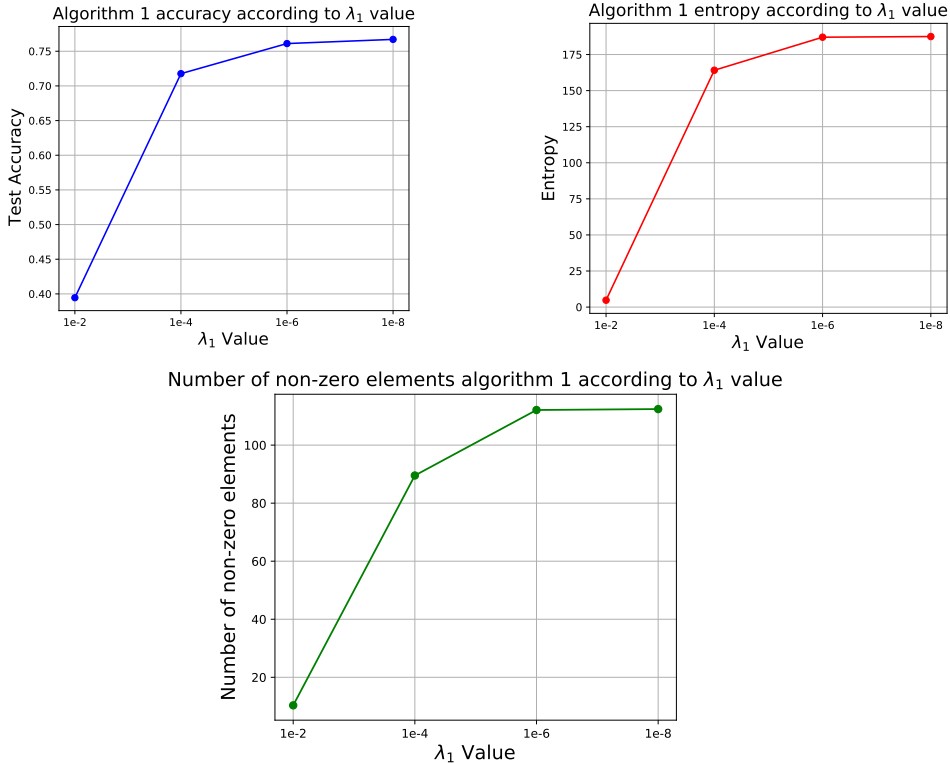

Figure 3: Classification accuracy and sparsity of local updates depending on $\lambda_1$ (Algorithm 1).

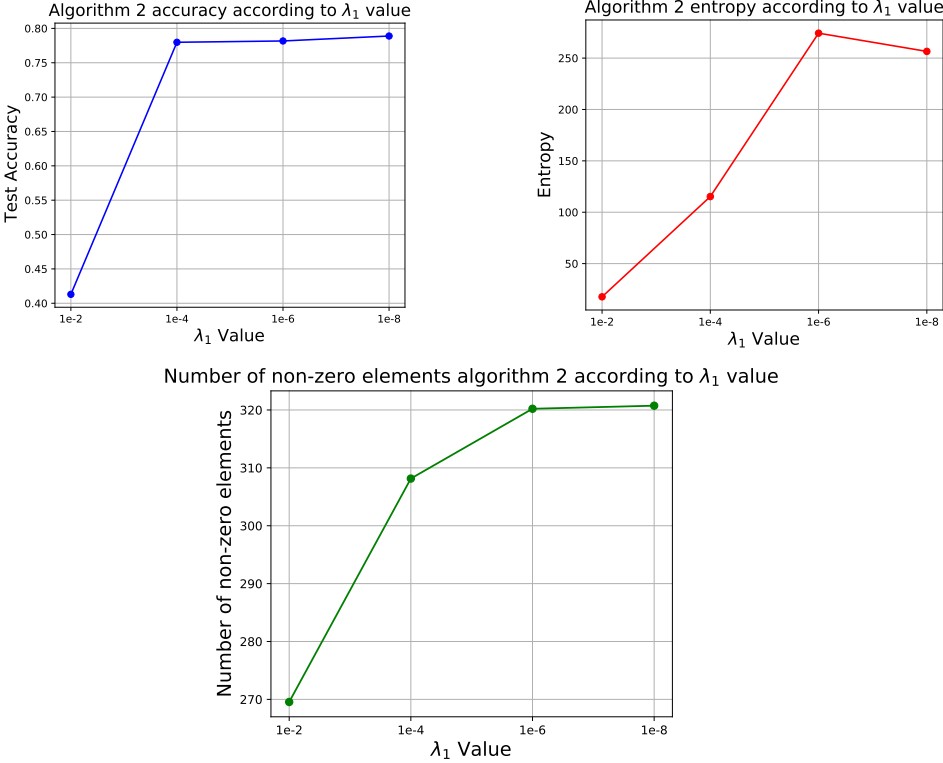

Figure 4: Classification accuracy and sparsity of local updates depending on $\lambda_1$ (Algorithm 2).

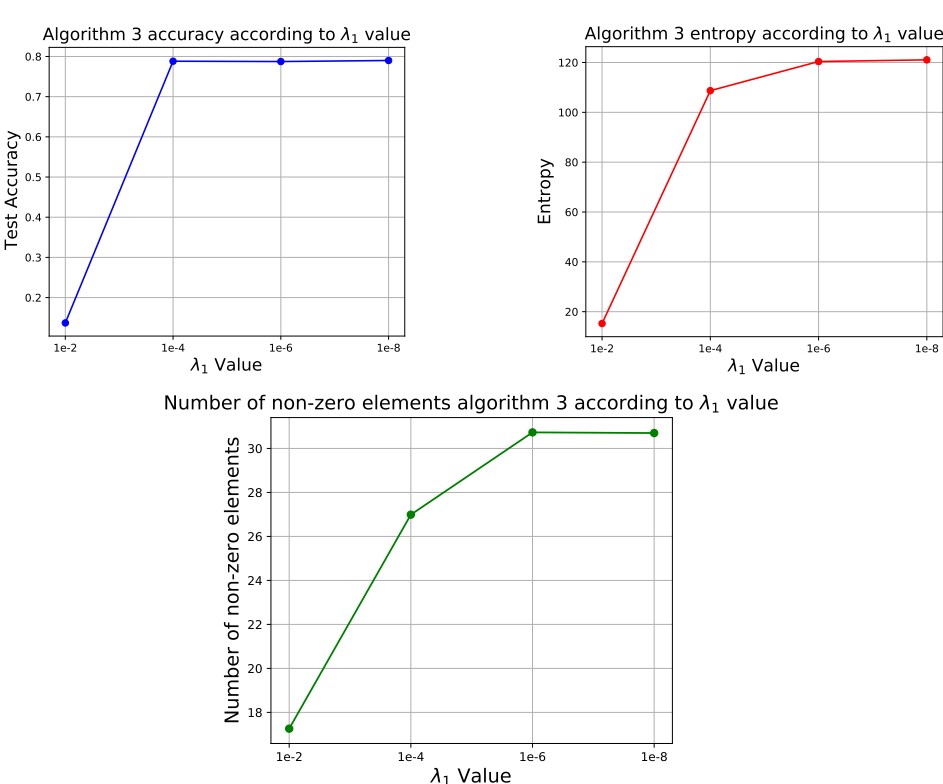

Figure 5: Classification accuracy and sparsity of local updates depending on $\lambda_1$ (Algorithm 3).

## A.3 EMPIRICAL RESULTS OF CLASSIFICATION ACCURACY

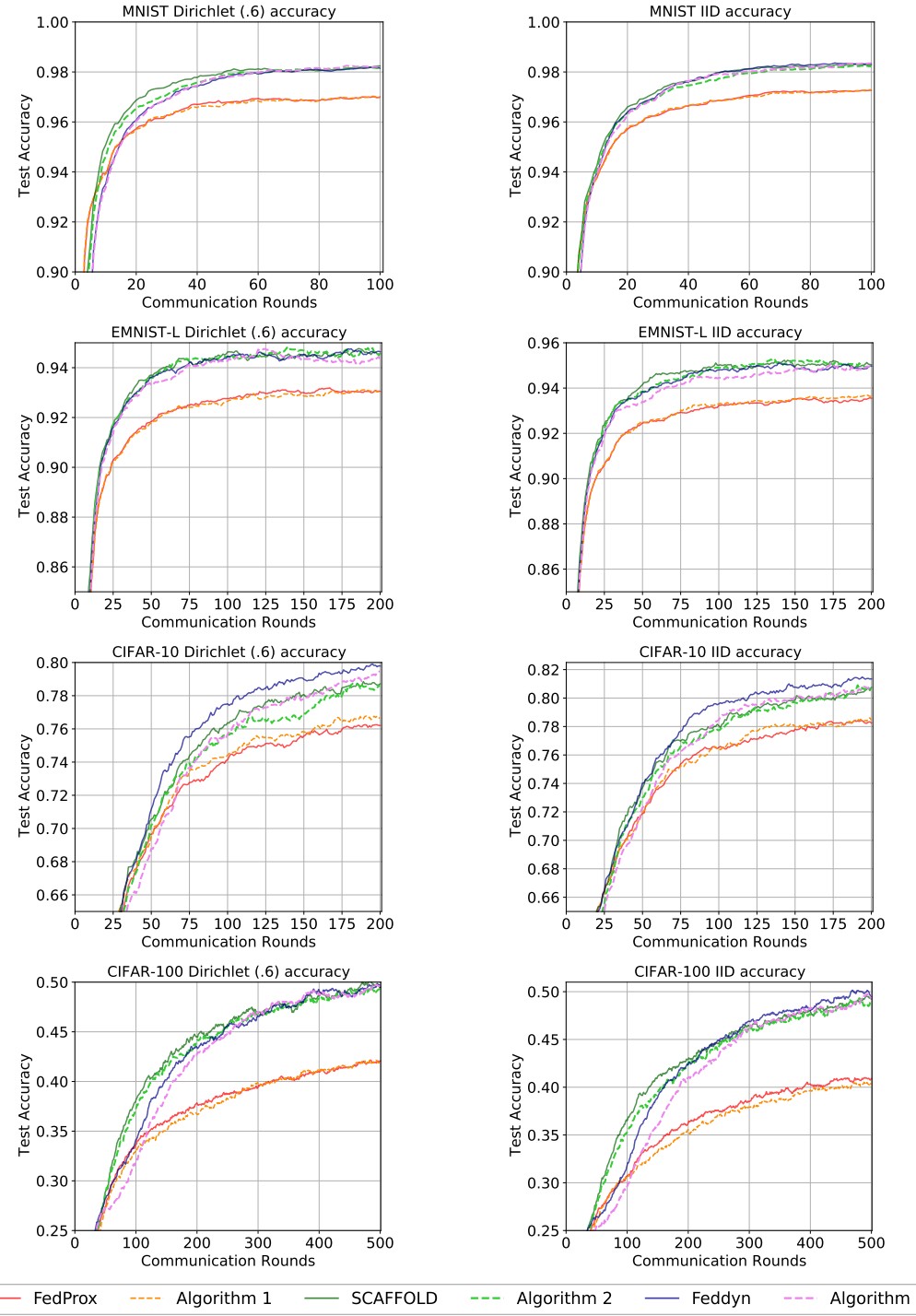

Figure 6: Classification accuracy performance evaluated in MNIST, EMNIST-L, CIFAR-10, and CIFAR-100 datasets (10% participation rate).

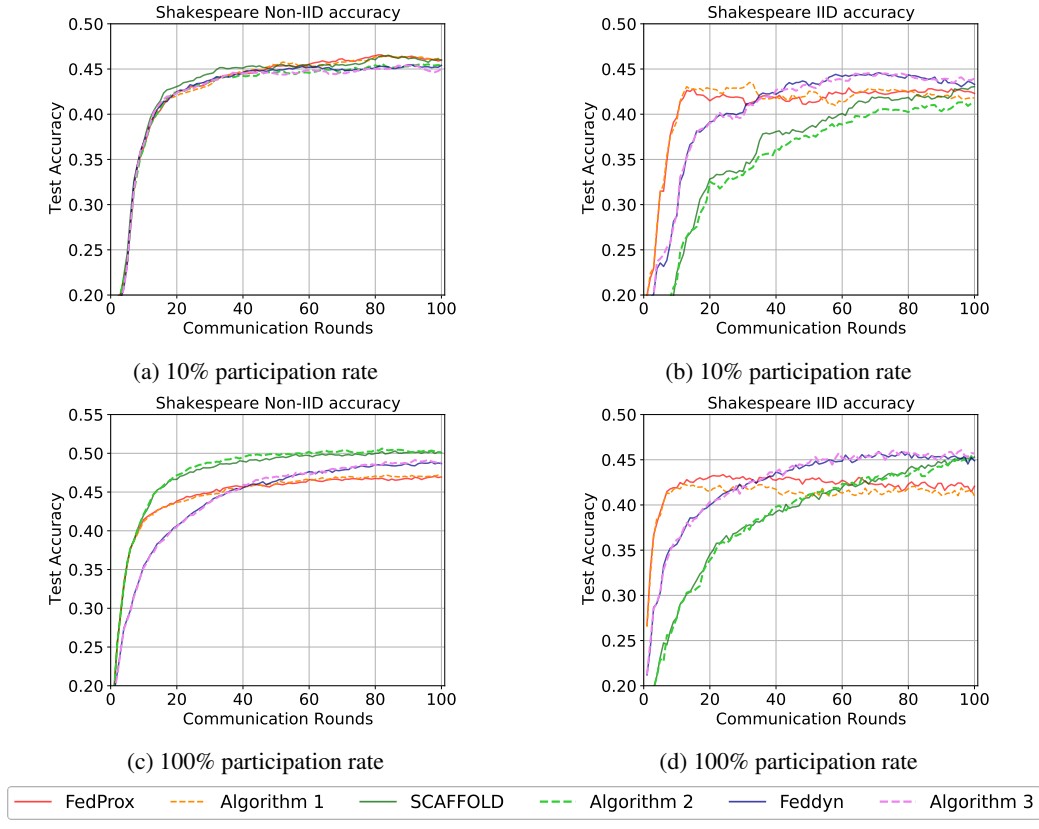

(a) 10% participation rate

(b) 10% participation rate

(c) 100% participation rate

(d) 100% participation rate

Figure 7: Classification accuracy performance evaluated in IID Shakespeare and Non-IID Shakespeare datasets.

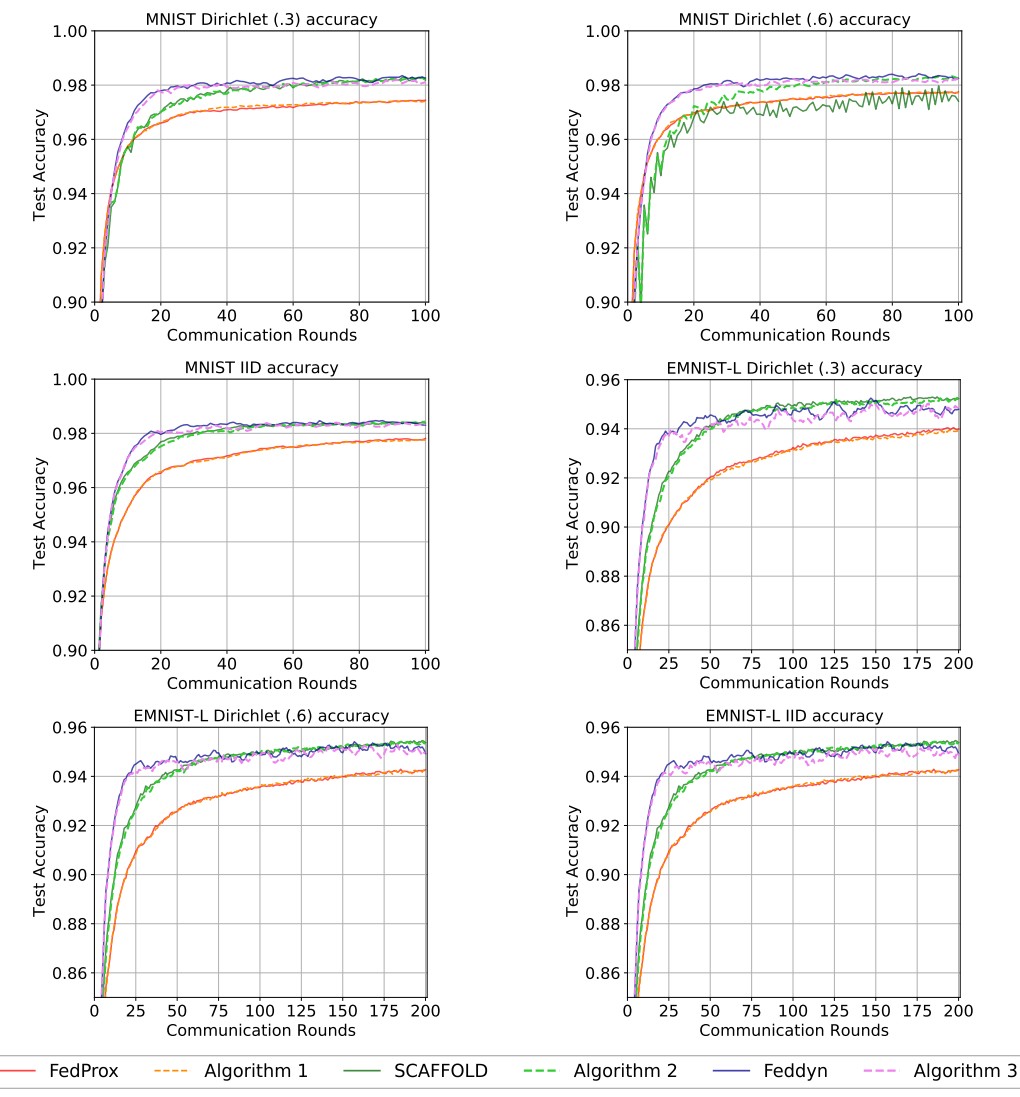

Figure 8: Classification accuracy performance evaluated in MNIST, EMNIST-L datasets (100% participation rate).

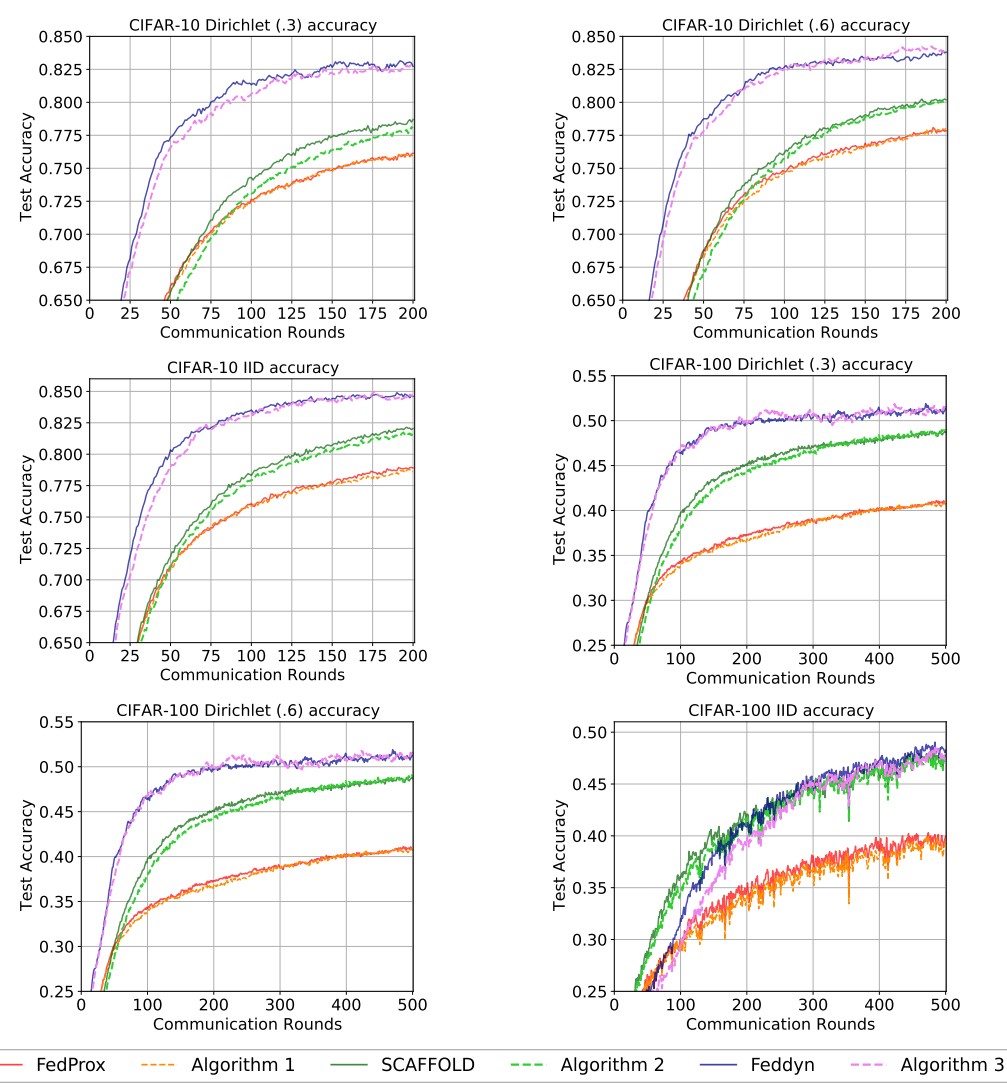

Figure 9: Classification accuracy performance evaluated in CIFAR-10 and CIFAR-100 datasets (100% participation rate).

# B PROOF

We utilize some techniques in FedDyn (Acar et al., 2021).

## B.1 DEFINITION

We introduce a formal definition and properties that we will use.

**Definition B.0.1.** *A function $L_k$ is $\beta$-smooth if it satisfies*

$$\|\nabla L_k(x) - \nabla L_k(y)\| \leq \beta \|x - y\| \quad \forall x, y. \tag{14}$$

If function $L_k$ is convex and $\beta$-smooth, it satisfies

$$-\langle \nabla L_k(x), z - y \rangle \leq -L_k(z) + L_k(y) + \frac{\beta}{2} \|z - x\|^2 \quad \forall x, y, z. \tag{15}$$

As a consequence of the convexity and smoothness, the following property holds (Nesterov, 2018, Theorem 2.1.5):

$$\frac{1}{2\beta m} \sum_{k \in [m]} \|\nabla L_k(x) - \nabla L_k(x_*)\|^2 \leq \mathcal{R}(x) - \mathcal{R}(x_*) \quad \forall x \tag{16}$$

where $\mathcal{R}(x) = \frac{1}{m} \sum_{k=1}^{m} L_k(x)$ and $\nabla \mathcal{R}(x_*) = 0$.

We will also use the relaxed triangle inequality (Karimireddy et al., 2020, Lemma 3):

$$\left\| \sum_{j=1}^{n} v_j \right\|^2 \leq n \sum_{j=1}^{n} \|v_j\|^2. \tag{17}$$

## B.2 PROOF OF THEOREM 3.1

The theorem that we will prove is as follows.

**Theorem B.1** (Full statement of Theorem 3.1). *Assume that the clients are uniformly randomly selected at each round and the individual loss functions $\{L_k\}_{k=1}^{m}$ are convex and $\beta$-smooth. Also assume that $\lambda_2 > 27\beta$. Then Algorithm 3 satisfies the following inequality: Letting $\mathcal{R}(\theta) = \frac{1}{m} \sum_{k \in [m]} L_k(\theta)$ and $\theta_* = \underset{\theta}{\arg\min} \mathcal{R}(\theta)$,*

$$\mathbb{E}\left[ \mathcal{R}\left( \frac{1}{T} \sum_{t=0}^{T-1} \gamma^t \right) - \mathcal{R}(\theta_*) \right] \leq \frac{1}{T} \frac{1}{\kappa_0} (\mathbb{E}\|\gamma^0 - \theta_*\|^2 + \kappa C_0) + \frac{\kappa'}{\kappa_0} \cdot \lambda_1^2 d$$

$$- \frac{1}{T} \frac{2\lambda_1}{\lambda_2} \sum_{t=1}^{T} \left\langle (\gamma^{t-1} - \theta_*), \frac{1}{m} \sum_{k \in [m]} \mathbb{E}[\operatorname{sign}(\tilde{\theta}_k^t - \theta^{t-1})] \right\rangle, \tag{18}$$

*where*

$$\gamma^t = \frac{1}{P} \sum_{k \in \mathcal{P}_t} \theta_k^t = \theta^t + \frac{1}{\lambda_2} h^t \quad \text{with } P = |\mathcal{P}_t|,$$

$$\kappa = \frac{10m}{P} \frac{1}{\lambda_2} \frac{\lambda_2 + \beta}{\lambda_2^2 - 25\beta^2},$$

$$\kappa_0 = \frac{2}{\lambda_2} \frac{\lambda_2^2 - 25\lambda_2\beta - 50\beta^2}{\lambda_2^2 - 25\beta^2},$$

$$\kappa' = \frac{5}{\lambda_2} \frac{\lambda_2 + \beta}{\lambda_2^2 - 25\beta^2} = \kappa \cdot \frac{P}{2m},$$

$$C_0 = \frac{1}{m} \sum_{k \in [m]} \mathbb{E}\|\nabla L_k(\theta_k^0) - \nabla L_k(\theta_*)\|,$$

$$d = dim(\theta).$$

To prove the theorem, define variables that will be used throughout the proof.

$$\tilde{\theta}_k^t = \arg\min_\theta L_k(\theta) - \langle \nabla L_k(\theta_k^{t-1}), \theta \rangle + \frac{\lambda_2}{2} \left\| \theta - \theta^{t-1} \right\|_2^2 + \lambda_1 \left\| \theta - \theta^{t-1} \right\|_1 \quad \forall k \in [m] \quad (19)$$

$$C_t = \frac{1}{m} \sum_{k \in [m]} \mathbb{E} \| \nabla L_k(\theta_k^t) - \nabla L_k(\theta_*) \|^2, \tag{20}$$

$$\epsilon_t = \frac{1}{m} \sum_{k \in [m]} \mathbb{E} \| \tilde{\theta}_k^t - \gamma^{t-1} \|^2. \tag{21}$$

Note that $\tilde{\theta}_k^t$ optimizes the $k$th loss function by assuming that the $k$th client ($k \in [m]$) is selected at round $t$. It is obvious that $\tilde{\theta}_k^t = \theta_k^t$ if $k \in \mathcal{P}_t$. $C_t$ refers to the average of the expected differences between gradients of each individual model and the globally optimal model. Lastly, $\epsilon_t$ refers to the deviation of each client model from the average of local models. Remark that $C_t$ and $\epsilon_t$ approach zero if all clients' models converge to the globally optimal model, i.e., $\theta_k^t \to \theta_*$.

The following lemma expresses $h^t$, how much the averaged active devices' model deviates from the global model.

**Lemma B.2.** *Algorithm 3 satisfies*

$$h^t = \frac{1}{m} \sum_{k \in [m]} \nabla L_k(\theta_k^t) \tag{22}$$

*Proof.* Starting from the update of $h^t$ in Algorithm 3,

$$h^t = h^{t-1} - \frac{\lambda_2}{m} \sum_{k \in [m]} (\theta_k^t - \theta^{t-1}) - \frac{\lambda_1}{m} \sum_{k \in [m]} \text{sign}(\theta_k^t - \theta^{t-1})$$

$$= h^{t-1} - \frac{1}{m} \sum_{k \in [m]} (\nabla L_k(\theta_k^{t-1}) - \nabla L_k(\theta_k^t) - \lambda_1 \text{sign}(\theta_k^t - \theta^{t-1})) - \frac{\lambda_1}{m} \sum_{k \in [m]} \text{sign}(\theta_k^t - \theta^{t-1})$$

$$= h^{t-1} - \frac{1}{m} \sum_{k \in [m]} (\nabla L_k(\theta_k^{t-1}) - \nabla L_k(\theta_k^t)),$$

where the second equality follows from (11). By summing $h^t$ recursively, we have

$$h^t = h^0 + \frac{1}{m} \sum_{k \in [m]} \nabla L_k(\theta_k^t) - \frac{1}{m} \sum_{k \in [m]} \nabla L_k(\theta_k^0) = \frac{1}{m} \sum_{k \in [m]} \nabla L_k(\theta_k^t).$$

$\square$

The next lemma provides how much the average of local models changes by using only $t$ round parameters.

**Lemma B.3.** *Algorithm 3 satisfies*

$$\mathbb{E}[\gamma^t - \gamma^{t-1}] = \frac{1}{\lambda_2 m} \sum_{k \in [m]} \mathbb{E}[-\nabla L_k(\tilde{\theta}_k^t)] - \frac{\lambda_1}{\lambda_2 m} \sum_{k \in [m]} \mathbb{E}[\text{sign}(\tilde{\theta}_k^t - \theta^{t-1})].$$

*Proof.* Starting from the definition of $\gamma^t$,

$$
\begin{aligned}
\mathbb{E}\left[\gamma^t - \gamma^{t-1}\right] &= \mathbb{E}\left[\left(\frac{1}{P}\sum_{k\in\mathcal{P}_t}\theta_k^t\right) - \theta^{t-1} - \frac{1}{\lambda_2}h^{t-1}\right] \\
&= \mathbb{E}\left[\frac{1}{P}\sum_{k\in\mathcal{P}_t}(\theta_k^t - \theta^{t-1}) - \frac{1}{\lambda_2}h^{t-1}\right] \\
&= \mathbb{E}\left[\frac{1}{\lambda_2 P}\sum_{k\in\mathcal{P}_t}(\nabla L_k(\theta_k^{t-1}) - \nabla L_k(\theta_k^t) - \lambda_1\mathrm{sign}(\theta_k^t - \theta^{t-1})) - \frac{1}{\lambda_2}h^{t-1}\right] \quad (23) \\
&= \mathbb{E}\left[\frac{1}{\lambda_2 P}\sum_{k\in\mathcal{P}_t}(\nabla L_k(\theta_k^{t-1}) - \nabla L_k(\tilde\theta_k^t) - \lambda_1\mathrm{sign}(\tilde\theta_k^t - \theta^{t-1})) - \frac{1}{\lambda_2}h^{t-1}\right] \quad (24) \\
&= \mathbb{E}\left[\frac{1}{\lambda_2 m}\sum_{k\in[m]}(\nabla L_k(\theta_k^{t-1}) - \nabla L_k(\tilde\theta_k^t) - \lambda_1\mathrm{sign}(\tilde\theta_k^t - \theta^{t-1})) - \frac{1}{\lambda_2}h^{t-1}\right] \\
&\hspace{11cm} (25) \\
&= \frac{1}{\lambda_2 m}\sum_{k\in[m]}\mathbb{E}[-\nabla L_k(\tilde\theta_k^t)] - \frac{\lambda_1}{\lambda_2 m}\sum_{k\in[m]}\mathbb{E}[\mathrm{sign}(\tilde\theta_k^t - \theta^{t-1})], \quad (26)
\end{aligned}
$$

where (23) follows from (11), (24) follows since $\tilde\theta_k^t = \theta_k^t$ if $k \in \mathcal{P}_t$, and (25) follows since clients are randomly chosen. The last equality is due to Lemma B.2. $\qquad\square$

Next, note that Algorithm 3 is the same as that of FedDyn except for the $\ell_1$-norm penalty. As this new penalty does not affect derivations of $C_t$, $\epsilon_t$, and $\mathbb{E}\|\gamma^t - \gamma^{t-1}\|^2$ in FedDyn (Acar et al., 2021), we can obtain the following bounds on them. Proofs are omitted for brevity.

$$
\mathbb{E}\|h^t\|^2 \le C_t \tag{27}
$$

$$
C_t \le \left(1 - \frac{P}{m}\right)C_{t-1} + \frac{2\beta^2 P}{m}\epsilon_t + \frac{4\beta P}{m}\mathbb{E}[\mathcal{R}(\gamma^{t-1}) - \mathcal{R}(\theta_*)] \tag{28}
$$

$$
\mathbb{E}\|\gamma^t - \gamma^{t-1}\|^2 \le \frac{1}{m}\sum_{k\in[m]}\mathbb{E}[\|\tilde\theta_k^t - \gamma^{t-1}\|^2] = \epsilon_t \tag{29}
$$

**Lemma B.4.** *Given model parameters at the round $(t-1)$, Algorithm 3 satisfies*

$$
\mathbb{E}\|\gamma^t - \theta_*\|^2 \le \mathbb{E}\|\gamma^{t-1} - \theta_*\|^2 - \frac{2}{\lambda_2}\mathbb{E}[\mathcal{R}(\gamma^{t-1}) - \mathcal{R}(\theta_*)] + \frac{\beta}{\lambda_2}\epsilon_t + \mathbb{E}\|\gamma^t - \gamma^{t-1}\|^2 \tag{30}
$$

$$
- \frac{2\lambda_1}{\lambda_2 m}(\gamma^{t-1} - \theta_*)\sum_{k\in[m]}\mathbb{E}[sign(\tilde\theta_k^t - \theta^{t-1})], \tag{31}
$$

*where the expectations are taken assuming parameters at the round $(t-1)$ are given.*

*Proof.*

$$
\begin{aligned}
\mathbb{E}\|\gamma^t - \theta_*\|^2 &= \mathbb{E}\|\gamma^{t-1} - \theta_* + \gamma^t - \gamma^{t-1}\|^2 \\
&= \mathbb{E}\|\gamma^{t-1} - \theta_*\|^2 + 2\mathbb{E}[\langle \gamma^{t-1} - \theta_*, \gamma^t - \gamma^{t-1}\rangle] + \mathbb{E}\|\gamma^t - \gamma^{t-1}\|^2 \\
&= \mathbb{E}\|\gamma^{t-1} - \theta_*\|^2 + \mathbb{E}\|\gamma^t - \gamma^{t-1}\|^2 \\
&\quad + \frac{2}{\lambda_2 m} \sum_{k\in[m]} \mathbb{E}\left[\left\langle \gamma^{t-1} - \theta_*, -\nabla L_k(\tilde\theta_k^t) - \lambda_1(\mathrm{sign}(\tilde\theta_k^t - \theta^{t-1}))\right\rangle\right] \quad (32) \\
&\leq \mathbb{E}\|\gamma^{t-1} - \theta_*\|^2 + \mathbb{E}\|\gamma^t - \gamma^{t-1}\|^2 \\
&\quad + \frac{2}{\lambda_2 m} \sum_{k\in[m]} \mathbb{E}[L_k(\theta_*) - L_k(\gamma^{t-1}) + \frac{\beta}{2}\|\tilde\theta_k^t - \gamma^{t-1}\|^2] \\
&\quad + \frac{2}{\lambda_2 m} \sum_{k\in[m]} \mathbb{E}\left[\left\langle \gamma^{t-1} - \theta_*, -\lambda_1\mathrm{sign}(\tilde\theta_k^t - \theta^{t-1})\right\rangle\right] \quad (33) \\
&= \mathbb{E}\|\gamma^{t-1} - \theta_*\|^2 + \mathbb{E}\|\gamma^t - \gamma^{t-1}\|^2 - \frac{2}{\lambda_2}\mathbb{E}[\mathcal{R}(\gamma^{t-1}) - \mathcal{R}(\theta_*)] + \frac{\beta}{\lambda_2}\epsilon_t \\
&\quad - \frac{2\lambda_1}{\lambda_2 m} \sum_{k\in[m]} \mathbb{E}\left[\left\langle \gamma^{t-1} - \theta_*, \mathrm{sign}(\tilde\theta_k^t - \theta^{t-1})\right\rangle\right] \quad (34) \\
&= \mathbb{E}\|\gamma^{t-1} - \theta_*\|^2 + \mathbb{E}\|\gamma^t - \gamma^{t-1}\|^2 - \frac{2}{\lambda_2}\mathbb{E}[\mathcal{R}(\gamma^{t-1}) - \mathcal{R}(\theta_*)] + \frac{\beta}{\lambda_2}\epsilon_t \\
&\quad - \frac{2\lambda_1}{\lambda_2}\left\langle \gamma^{t-1} - \theta_*, \frac{1}{m}\sum_{k\in[m]} \mathbb{E}[\mathrm{sign}(\tilde\theta_k^t - \theta^{t-1})]\right\rangle \quad (35)
\end{aligned}
$$

where (32) follows from Lemma B.3, (33) follows from (15), and (34) follows from the definitions of $\mathcal{R}(\cdot)$ and $\epsilon_t$. $\qquad\square$

**Lemma B.5.** *Algorithm 3 satisfies*

$$
(1 - 5\frac{\beta^2}{\lambda_2^2})\epsilon_t \leq 10\frac{1}{\lambda_2^2}C_{t-1} + 10\beta\frac{1}{\lambda_2^2}\mathbb{E}[\mathcal{R}(\gamma^{t-1}) - \mathcal{R}(\theta_*)] + \frac{5\lambda_1^2}{\lambda_2^2}d
$$

*Proof.* Starting from the definitions of $\epsilon_t$ and $\gamma^t$,

$$
\epsilon_t = \frac{1}{m} \sum_{k \in [m]} \mathbb{E}\|\tilde{\theta}_k^t - \gamma^{t-1}\|^2
$$

$$
= \frac{1}{m} \sum_{k \in [m]} \mathbb{E}\|\tilde{\theta}_k^t - \theta^{t-1} - \frac{1}{\lambda_2}h^{t-1}\|^2
$$

$$
= \frac{1}{\lambda_2^2} \frac{1}{m} \sum_{k \in [m]} \mathbb{E}\|\nabla L_k(\theta_k^{t-1}) - \nabla L_k(\tilde{\theta}_k^t) - \lambda_1 \mathrm{sign}(\theta_k^t - \theta^{t-1}) - h^{t-1}\|^2 \tag{36}
$$

$$
= \frac{1}{\lambda_2^2} \frac{1}{m} \sum_{k \in [m]} \mathbb{E}\|\nabla L_k(\theta_k^{t-1}) - \nabla L_k(\theta_*) + \nabla L_k(\theta_*) - \nabla L_k(\gamma^{t-1})
$$

$$
\qquad\qquad + \nabla L_k(\gamma^{t-1}) - \nabla L_k(\tilde{\theta}_k^t) - \lambda_1 \mathrm{sign}(\theta_k^t - \theta^{t-1}) - h^{t-1}\|^2
$$

$$
\leq \frac{5}{\lambda_2^2} \frac{1}{m} \sum_{k \in [m]} \mathbb{E}\|\nabla L_k(\theta_k^{t-1}) - \nabla L_k(\theta_*)\|^2 + \frac{5}{\lambda_2^2} \frac{1}{m} \sum_{k \in [m]} \mathbb{E}\|\nabla L_k(\gamma_k^{t-1}) - \nabla L_k(\theta_*)\|^2
$$

$$
+ \frac{5}{\lambda_2^2} \frac{1}{m} \sum_{k \in [m]} \mathbb{E}\|\nabla L_k(\tilde{\theta}_k^t) - \nabla L_k(\gamma^{t-1})\|^2 + \frac{5}{\lambda_2^2}\mathbb{E}\|\lambda_1 \mathrm{sign}(\theta_k^t - \theta^{t-1})\|^2 + \frac{5}{\lambda_2^2}\mathbb{E}\|h^{t-1}\|^2
$$
$$\tag{37}$$

$$
\leq \frac{5}{\lambda_2^2} \frac{1}{m} \sum_{k \in [m]} \mathbb{E}\|\nabla L_k(\theta_k^{t-1}) - \nabla L_k(\theta_*)\|^2 + \frac{5}{\lambda_2^2} \frac{1}{m} \sum_{k \in [m]} \mathbb{E}\|\nabla L_k(\gamma_k^{t-1}) - \nabla L_k(\theta_*)\|^2
$$

$$
\qquad\qquad + \frac{5}{\lambda_2^2} \frac{1}{m} \sum_{k \in [m]} \mathbb{E}\|\nabla L_k(\tilde{\theta}_k^t) - \nabla L_k(\gamma^{t-1})\|^2 + \frac{5\lambda_1^2}{\lambda_2^2}d + \frac{5}{\lambda_2^2}C_{t-1} \tag{38}
$$

$$
\leq \frac{5}{\lambda_2^2}C_{t-1} + \frac{5}{\lambda_2^2}2\beta\,\mathbb{E}[\mathcal{R}(\gamma^{t-1}) - \mathcal{R}(\theta_*)] + \frac{5\beta^2}{\lambda_2^2} \frac{1}{m} \sum_{k \in [m]} \mathbb{E}\|\tilde{\theta}_k^t - \gamma^{t-1}\|^2 + \frac{5\lambda_1^2}{\lambda_2^2}d + \frac{5}{\lambda_2^2}C_{t-1}
$$
$$\tag{39}$$

$$
= \frac{10}{\lambda_2^2}C_{t-1} + \frac{10\beta}{\lambda_2^2}\mathbb{E}[\mathcal{R}(\gamma^{t-1}) - \mathcal{R}(\theta_*)] + \frac{5\beta^2}{\lambda_2^2}\epsilon_t + \frac{5\lambda_1^2}{\lambda_2^2}d,
$$

where (36) follows from (11), (37) follows from the relaxed triangle inequality (17), (38) follows from (27), and (39) follows from the definition of $C_t$, the smoothness, and (16). The last equality follows from the definition of $\epsilon_t$. $\qquad\square$

After multiplying (28) by $\kappa(= 10\frac{m}{P}\frac{1}{\lambda_2}\frac{\lambda_2+\beta}{\lambda_2^2-25\beta^2})$, we obtain the following theorem by summing (B.4) and scaled version of (29).

**Theorem B.6.** *Given model parameters at the round $(t-1)$, Algorithm 3 satisfies*

$$
\kappa_0\mathbb{E}[\mathcal{R}(\gamma^{t-1}) - \mathcal{R}(\theta_*)] \leq (\mathbb{E}\|\gamma^{t-1} - \theta_*\|^2 + \kappa C_{t-1}) - (\mathbb{E}\|\gamma^t - \theta_*\|^2 + \kappa C_t) + \kappa\frac{P}{2m}\lambda_1^2
$$

$$
- \frac{2\lambda_1}{\lambda_2}\left\langle \gamma^{t-1} - \theta_*, \frac{1}{m}\sum_{k \in [m]}\mathbb{E}[\mathrm{sign}(\tilde{\theta}_k^t - \theta^{t-1})]\right\rangle.
$$

*where $\kappa = 10\frac{m}{P}\frac{1}{\lambda_2}\frac{\lambda_2+\beta}{\lambda_2^2-25\beta^2}, \kappa_0 = \frac{2}{\lambda_2}\frac{\lambda_2^2-25\lambda_2\beta-50\beta^2}{\lambda_2^2-25\beta^2}$. Note that the expectations taken above are conditional expectations given model parameters at time $(t-1)$.*

*Proof.* Summing Lemma $B.4$ and $\kappa$-scaled version of (28), we have

$$\mathbb{E}\|\gamma^t - \theta_*\|^2 + \kappa C_t$$

$$\leq \mathbb{E}\|\gamma^{t-1} - \theta_*\|^2 + \kappa C_{t-1} - \kappa \frac{P}{m} C_{t-1} + \kappa \frac{2\beta^2 P}{m}\epsilon_t + \kappa \frac{4\beta P}{m}\mathbb{E}[\mathcal{R}(\gamma^{t-1}) - \mathcal{R}(\theta_*)]$$

$$- \frac{2}{\lambda_2}\mathbb{E}[\mathcal{R}(\gamma^{t-1}) - \mathcal{R}(\theta_*)] + \frac{\beta}{\lambda_2}\epsilon_t + \mathbb{E}\|\gamma^t - \gamma^{t-1}\|^2 - \frac{2\lambda_1}{\lambda_2 m}(\gamma^{t-1} - \theta_*)\sum_{k \in [m]}\mathbb{E}[\text{sign}(\tilde{\theta}_k^t - \theta^{t-1})].$$

$$(40)$$

As $\mathbb{E}\|\gamma^t - \gamma^{t-1}\|^2 \leq \epsilon_t$ by (29), we have

$$\kappa \frac{2\beta^2 P}{m}\epsilon_t + \frac{\beta}{\lambda_2}\epsilon_t + \mathbb{E}\|\gamma^t - \gamma^{t-1}\|^2 \leq \kappa \frac{2\beta^2 P}{m}\epsilon_t + \frac{\beta}{\lambda_2}\epsilon_t + \epsilon_t. \qquad (41)$$

This can be further bounded as follows.

$$(41) = \left(10\frac{m}{P}\frac{1}{\lambda_2}\frac{\lambda_2 + \beta}{\lambda_2^2 - 25\beta^2} \cdot \frac{2\beta^2 P}{m} + \frac{\beta}{\lambda_2} + 1\right)\epsilon_t$$

$$= \frac{1}{\lambda_2(\lambda_2^2 - 25\beta^2)}\left(20(\lambda_2 + \beta)\beta^2 + \beta(\lambda_2^2 - 25\beta^2) + \lambda_2(\lambda_2^2 - 25\beta^2)\right)\epsilon_t$$

$$= \frac{\lambda_2(\lambda_2 + \beta)}{\lambda_2^2 - 25\beta^2}\left(1 - 5\frac{\beta^2}{\lambda_2^2}\right)\epsilon_t$$

$$\leq \frac{\lambda_2(\lambda_2 + \beta)}{\lambda_2^2 - 25\beta^2}\left(\frac{10}{\lambda_2^2}C_{t-1} + \frac{10\beta}{\lambda_2^2}\mathbb{E}[\mathcal{R}(\gamma^{t-1}) - \mathcal{R}(\theta_*)] + \frac{5\lambda_1^2}{\lambda_2^2}d\right)$$

$$= \kappa \frac{P}{m}C_{t-1} + \kappa \frac{\beta P}{m}\mathbb{E}[\mathcal{R}(\gamma^{t-1}) - \mathcal{R}(\theta_*)] + \kappa \frac{P}{2m}\lambda_1^2 d,$$

where the inequality follows from Lemma B.5. Then, (40) term will be

$$\mathbb{E}\|\gamma^t - \theta_*\|^2 + \kappa C_t \leq \mathbb{E}\|\gamma^{t-1} - \theta_*\|^2 + \kappa C_{t-1} - \kappa_0 \mathbb{E}[\mathcal{R}(\gamma^{t-1}) - \mathcal{R}(\theta_*)] + \kappa \frac{P}{2m}\lambda_1^2 d$$

$$- \frac{2\lambda_1}{\lambda_2}\left\langle\gamma^{t-1} - \theta_*, \frac{1}{m}\sum_{k \in [m]}\mathbb{E}[\text{sign}(\tilde{\theta}_k^t - \theta^{t-1})]\right\rangle.$$

Rearranging terms, we prove the claim. $\qquad\square$

Now we are ready to prove the main claim by combining all lemmas. Let us take the sum on both sides of Lemma B.6 over $t = 1, \ldots, T$. Then, telescoping gives us

$$\kappa_0 \sum_{t=1}^T \mathbb{E}[\mathcal{R}(\gamma^{t-1}) - \mathcal{R}(\theta_*)] \leq (\mathbb{E}\|\gamma^0 - \theta_*\|^2 + \kappa C_0) - (\mathbb{E}\|\gamma^T - \theta_*\|^2 + \kappa C_T) + T(\kappa \frac{P}{2m}\lambda_1^2)$$

$$- \frac{2\lambda_1}{\lambda_2}\sum_{t=1}^T \left\langle\gamma^{t-1} - \theta_*, \frac{1}{m}\sum_{k \in [m]}\mathbb{E}[\text{sign}(\tilde{\theta}_k^t - \theta^{t-1})]\right\rangle.$$

Since $\kappa$ is positive if $\lambda_2 > 27\beta$, we can eliminate the negative term in the middle. Then,

$$\kappa_0 \sum_{t=1}^T \mathbb{E}[\mathcal{R}(\gamma^{t-1}) - \mathcal{R}(\theta_*)] \leq \mathbb{E}\|\gamma^0 - \theta_*\|^2 + \kappa C_0 + T(\kappa \frac{P}{2m}\lambda_1^2 d)$$

$$- \frac{2\lambda_1}{\lambda_2}\sum_{t=1}^T \left\langle\gamma^{t-1} - \theta_*, \frac{1}{m}\sum_{k \in [m]}\mathbb{E}[\text{sign}(\tilde{\theta}_k^t - \theta^{t-1})]\right\rangle.$$

Dividing by $T$ and applying Jensen's inequality,

$$
\mathbb{E}\left[\mathcal{R}(\frac{1}{T}\sum_{t=0}^{T-1}\gamma^t) - \mathcal{R}(\theta_*)\right] \leq \frac{1}{T}\frac{1}{\kappa_0}(\mathbb{E}\|\gamma^0 - \theta_*\|^2 + \kappa C_0) + \frac{1}{\kappa_0}(\kappa\frac{P}{2m}\lambda_1^2 d)
$$
$$
- \frac{1}{T}\frac{2\lambda_1}{\lambda_2}\sum_{t=1}^{T}\left\langle \gamma^{t-1} - \theta_*, \frac{1}{m}\sum_{k\in[m]}\mathbb{E}[\text{sign}(\tilde{\theta}_k^t - \theta^{t-1})]\right\rangle,
$$
(42)

which completes the proof of Theorem B.1.

## B.3 DISCUSSION ON CONVERGENCE

In this section, we revisit the convergence stated in Theorem 3.1. Recall the bound

$$
\mathbb{E}\left[\mathcal{R}\left(\frac{1}{T}\sum_{t=0}^{T-1}\gamma^t\right) - \mathcal{R}(\theta_*)\right] \leq \frac{1}{T}\frac{1}{\kappa_0}(\mathbb{E}\|\gamma^0 - \theta_*\|^2 + \kappa C_0) + \frac{1}{\kappa_0}(\kappa\frac{P}{2m}\lambda_1^2 d)
$$
$$
- \frac{1}{T}\frac{2\lambda_1}{\lambda_2}\sum_{t=1}^{T}\left\langle \gamma^{t-1} - \theta_*, \frac{1}{m}\sum_{k\in[m]}\mathbb{E}[\text{sign}(\tilde{\theta}_k^t - \theta^{t-1})]\right\rangle,
$$

As we discussed in the main body, the second term is a negligible constant in the range of our hyperparameters as $\lambda_1$ is of order of $10^{-4}$ or $10^{-6}$.

Consider the last term where the summand is the inner product between two terms: 1) $\gamma^{t-1} - \theta_*$, the deviation of the averaged local models from the globally optimal model and 2) the average of sign vectors across clients. The deviation term characterizes how much the averaged local models are different from the global model; thus, we can assume that as training proceeds it vanishes or at least is bounded by a constant vector. To argue the average of sign vectors, assume a special case where the sign vectors $\text{sign}(\tilde{\theta}_k^t - \theta^{t-1})$ are IID across clients. To further simplify the argument, let us consider only a single coordinate of the sign vectors, say $X_k = \text{sign}(\tilde{\theta}_k^t(i) - \theta^{t-1}(i))$, and suppose $X_k = \pm 1$ with probability 0.5 each. Then, the concentration inequality (Durrett, 2019) implies that for any $\delta > 0$,

$$
\mathbb{P}\left[\frac{1}{m}\sum_{k\in[m]}\text{sign}(\tilde{\theta}_k^t) - \theta^{t-1} > \delta\right] = \mathbb{P}\left[\frac{1}{m}\sum_{k\in[m]}X_k > \delta\right] \leq e^{-\frac{m\delta^2}{2}}
$$

holds, which vanishes exponentially fast with the number of clients $m$. Since $m$ is large in many FL scenarios, the average of sign vectors is negligible with high probability, which in turn implies the last term is also negligible.

