# OpenReview forum: "Communication-Efficient and Drift-Robust Federated Learning via Elastic Net"
_ICLR.cc/2023/Conference — Submitted to ICLR 2023_

### Official Review · Reviewer_jTwe · 2022-10-19

**Confidence:** 4
**Correctness:** 4
**Technical Novelty And Significance:** 2
**Empirical Novelty And Significance:** 2
**Recommendation:** 3

**Clarity, Quality, Novelty And Reproducibility:**

The paper is very well written, and derivations are clear and easy to follow. Experimental setup is quite easy to follow.

**Strength And Weaknesses:**

Strength
- For one variant authors provide theoretical evidence that the augmentation does not harm convergence
Weakness
- Theory makes very strong assumptions
- Theory does not prove improvement over baseline method, nor does it help much with practical questions (selecting tuning parameters etc).
- Experiments show only marginal improvements over the baseline in terms of both compression and quality.

**Summary Of The Paper:**

The paper proposes adding an elastic net penalty while calculating client updates to the server model. The authors show that this augmentation can be implemented with a variety of standard FL optimization methods, provide theoretical convergence analysis for well behaved (convex, smooth) objective functions, and show empirically that adding this loss produces better results, both in terms of model update communication cost, and accuracy.

**Summary Of The Review:**

The work is accurate, but does not add sufficiently novel ideas. Utilizing elastic loss while training server side is already well established, and the loss function is well studied. Adding this loss to regularize the difference between the server model and client updated model is a fairly simple extension, and neither the theory nor experimental results are super exciting.

---

### Official Review · Reviewer_Rp9d · 2022-10-25

**Confidence:** 3
**Clarity, Quality, Novelty And Reproducibility:** The paper is readable, but the empiri…
**Correctness:** 2
**Technical Novelty And Significance:** 1
**Empirical Novelty And Significance:** 1
**Recommendation:** 3

**Strength And Weaknesses:**

Main concerns:
1. FedProx and FedDyn are also FL schemes that utilize regularization ($\ell_2$-like), thus this paper adds the $\ell_1$ term with a sign adjustment.
2. The last sentence in page 6: "We optimize the hyperparameters depending on the evaluated dataset: learning rates, $\lambda_2$, $\lambda_1$" is dubious as it relates to novelty since it can come down to the task of tuning these parameters to achieve the desired result, which is acceptable for application.
3. Table 2 shows that the number of non-zero elements are reduced across the board, but it does not show "when" the non-zero elements become zero. This can be shown by a plot of non-zero elements vs communication round.

**Summary Of The Paper:**

The papers shows how to apply elastic net [Zou & Hastie 2005] to existing federated learning (FL) schemes. Specifically, three FL schemes are selected: FedProx (already incorporates $\ell_2$ regularization) [Li et al. 2020], SCAFFOLD [Karimireddy et al. 2020], and FedDyn [DAE Acar et al. 2021. At a high-level, elastic net is a regularization technique that combines $\ell_1$ and $\ell_2$ regularization.

The paper supports the claim that applying elastic net to the aforementioned FL schemes resolves the client drift and communication cost problem, in one fell swoop, through the following empirical evaluations:

1. Sparsity (communication cost): Table 2: "number of non-zero elements cumulated (accumulated?) over all round simulated with 10% client participation for IID and non-IID settings in FL scenarios."

2. Entropy (client drift): Table 3 and Table 4: "cumulative entropy values of transmitted bits with 10%" and "100% client participation for IID and non-IID settings in FL scenarios."

Additionally, test accuracy, as a function of communication rounds, results are also provided.

Furthermore, the paper utilizes a distribution plot (Figure 2 -- FedDyn, elastic net applied to SCAFFOLD & FedDyn), and reference to the text by Cover & Thomas 2006 to substantiate the claim that "FedElasticNet can reduce the entropy by transforming the Gaussian distribution into the non-Gaussian one."




**Summary Of The Review:**

The novelty of this paper is scarce since the main idea is to show how to use elastic net under a limited set of FL schemes. The claims in the paper are also informal.

---

### Official Review · Reviewer_odbQ · 2022-11-04

**Confidence:** 4
**Clarity, Quality, Novelty And Reproducibility:** The paper is clearly written, the ide…
**Correctness:** 2
**Technical Novelty And Significance:** 3
**Empirical Novelty And Significance:** 2
**Recommendation:** 3

**Strength And Weaknesses:**

#### Strengths
1. The idea is simple and intuitive.
2. The paper is overall well written and motivated.
3. The authors show $O(\frac{1}{T})$ convergence in the smooth, convex setting

#### Concerns
1. Table 1 - what do these symbols mean ? Please update the caption for readability.
2. Theorem 3.1 The second term depends on d -  will it be negligible when we are dealing with practical large models ? A supporting plot simulating the convergence rates would be helpful.

*Client Drift*

3. Results on non-IID: One claim of the paper is that it can effectively deal with both client drift and communication compression. However, non-IID exp is extremely limited, only Shakesepeare and one non-iid setting; To support the drift robust claims - there needs to be more experiments on multiple dataset , models, and across different strength of heterogeneity comparing with multiple method that only deal with client drift / data heterogeneity.

4. In the non-iid setting, SCAFFOLD which is a standard approach to deal with client drift seems to perform equally well or even better than the proposed solution; Can you add more discussion on this and what are the scenarios when SCAFFOLD is better and when the proposed algorithm is better.

5. The theory is based off FedDyn. Now, FedDyn and SCAFFOLD already deal with the client drift / heterogeneity and thus it is hard to flesh out the roles of the regularization penalty introduces in this paper.

6. Analysis on FedAvg + $\ell_1$ + $\ell_2$ i.e. Algorithm 1 needs to be done - to clearly show if there is any advantage from these penalty terms.

*Communication Compression*

7. One effective way to deal with communication compression is to use contractive compression operators ex. quantization, top-k etc. To deal with the slow convergence due to additional terms appearing in the convergence result due to compression, it is standard practice to use an error feedback mechanism that ensures linear convergence at the same rate. One can simply use any of the drift robust algorithm (SCAFFOLD e.g.) apply communication compression with EF and be communication efficient.
For fair comparison, you need to compare with different drift robust algo + compression + EF with the proposed solution.
Empirically, for different compression rate ( using Top-k, Sign, Quantization etc compressor + Error Feedback compression operation C ) a clean experiment is to compare : FedAvg + C + EF vs FedAvg + $\ell_1$ + $\ell_2$

8. Also in theory, it is beneficial to discuss the convergence rate obtained (additional terms) with that in case of EF + Compression (Quantize, top k , q sparse etc several available approaches )



**Summary Of The Paper:**

Federated Learning trains a global model collaboratively over a set of clients , where the data is kept locally at the clients and only local gradients or parameters are communicated periodically with the server. Communicating large number of bits mights often slow down convergence and thus various compression schemes are used to deal with communication latency. Another challenge in the federated framework is heterogenous data distribution across clients.

This paper tries to tackle both these issues. In particular, they propose FedElasticNet a new framework for communication- efficient and drift-robust FL. They exploit $\ell_1, \ell_2$ regularization to deal with communication efficiency ( since $\ell_1$ regularizer ensures sparse solution ) and client drift respectively.

**Summary Of The Review:**

Overall, while I feel the idea is nice - the paper in its current state lacks to establish the proposed claims -
Empirically we need ablations on:
( a ) How does it compare with existing drift robust algorithms
( b ) How does it compare with  existing algorithms that can deal with data heterogeneity.
( c ) No mention of compression + EF in the paper - which is the standard practice in any FL setting. Why would I use this method and not C + EF ?

---

### Official Review · Reviewer_9ArF · 2022-11-04

**Confidence:** 4
**Clarity, Quality, Novelty And Reproducibility:** please see above
**Correctness:** 2
**Technical Novelty And Significance:** 1
**Empirical Novelty And Significance:** 2
**Recommendation:** 3

**Strength And Weaknesses:**

Strength:
1. The topic of improving the efficiency and reducing client shift in FL is important.

Weakness:
1. The novelty is not significant. Both $l_1$ and $l_2$ regularizations are mature techniques in statistics, and deep neural network training. I do not find this method very interesting. Using Lasso to get sparsity is doable, but is very tricky to tune and leads to biased gradient estimation which would hurt the model performance. Indeed, the theory (Theorem 3.1) containg a nonvanishing constant is worse than the baseline FL rate, and emprical performance might also be worse in some cases as we see from the figures. Thus, the proposed method seems not very valuable both theoretically and empirically. Also, elastic net introduces two more hyper parameters $\lambda_1$ and $\lambda_2$ which make the method harder to tune. Thus, much stronger motivation is needed.

2. The related work section should be improved. The paper misses many recent papers on communication-efficient FL.

3. The presentation is not satisfactory.

(i) In Algorithm 1, why is the local update presented in this way? How do you solve that optimization problem locally? In practice we may use SGD as in Algorithm 2? Why are they inconsistent? Are we using stochastic optimziation?
(ii) In the theory part, there is no formal statement of the assumptions. I do not know the setting of this analysis. Particularly, which part is related to the client drift? The assumptions should be stated very clearly.


**Summary Of The Paper:**

The paper proposes FedElasticNet which applies elastic net to federated learning (FL) of deep neural nets. The paper proposes two variants based on FedAvg and FedDyn, and provides the convergence analysis of the latter. Experiments are conducted to show the benefits of the proposed methods.

**Summary Of The Review:**

In general, the paper is not very well written, and the motivation and performance of the proposed algorithm is poor. Thus, I think the paper does not meet the bar of ICLR.

---

### Decision · Program_Chairs · 2023-01-20

**Decision:**

Reject

**Justification For Why Not Higher Score:**

The novelty is limited. The experiment and theory are not interesting enough.

**Justification For Why Not Lower Score:**

N/A

**Metareview: Summary, Strengths And Weaknesses:**

This paper applied the well-known elastic net method for federate learning, with standard optimization procedure. While reviewers agree that the paper reported some useful results, they consider the novelty is not sufficient given the rich related literature on elastic net loss for training sever side. The extension is not novel enough and the experiments as well the theory are not interesting enough for reviewers to recommend an acceptance.